🔓 | **Open Peer Review** | Microbial Genetics | Research Article

# Comparison of Illumina and Oxford Nanopore Technology systems for the genomic characterization of *Streptococcus pneumoniae*

Fatima Dakroub,[1] Fata Akl,[1] Alissar Zaghlout,[1,2] Jose Rita Gerges,[1,2] Nancy Hourani,[1] Celina F. Boutros,[1] George F. Araj,[1,3] Ghassan M. Matar,[1,2] Antoine Abou Fayad,[1,2] Ghassan S. Dbaibo,[1,4] for the Lebanese Inter-Hospital Pneumococcal Surveillance Program investigators

**ABSTRACT** Whole-genome sequencing (WGS) is an invaluable tool that enables high-resolution genotyping to precisely identify bacterial strains. It is particularly significant for highly pathogenic bacteria such as *Streptococcus pneumoniae*, a worldwide leading cause of mortality and morbidity. Illumina sequencing is highly established for *S. pneumoniae*, while Oxford Nanopore Technologies (ONT) data are limited. Hence, evaluating ONT-only data is needed. We aimed to compare the Illumina and ONT systems for *S. pneumoniae* sequencing. Moreover, we aimed to explore whether the newer chemistry from ONT with R10.4.1 flow cells improves the data outputs from long-read sequencing. *S. pneumoniae* bacteria were isolated from hospitalized patients with invasive pneumococcal disease (IPD) and serotyped by multiplex PCR. Resistance profiles were determined with anti-microbial susceptibility testing. A total of 27 isolates were sequenced using ONT Mk1c with R9.4.1 flow cells and Kit10 chemistry (ONT_V10) and the Illumina Miseq system. Illumina and ONT data were compared, and hybrid assembly was assessed. ONT sequencing was additionally performed with R10.4.1 flow cells and Kit14 chemistry (ONT_V14) in 12 isolates. *S. pneumoniae* identification, serotyping, AMR, and GPSC prediction were successfully achieved using ONT sequencing. The ONT_V14 chemistry significantly improved both MLST and pbp prediction in long-read sequencing. Overall, the hybrid assembly produced circular and contiguous genomes with high N50 parameters. Moreover, long-read assembly followed by short-read polishing is a fast and reliable approach for hybrid assembly at ONT sequencing depth >100×. For ONT sequencing depth <50×, tools that perform short-read-first assembly, such as Unicycler are recommended.

**IMPORTANCE** This study provides a detailed evaluation of whole-genome sequencing technologies and bioinformatics pipelines for the characterization of *Streptococcus pneumoniae*. It represents an in-depth investigation of Illumina and Oxford Nanopore technologies (ONT) systems for bacterial sequencing. It sheds light on the performance of each platform in various aspects of sequencing, including raw and assembly statistics, capsular typing, pbp typing, GPSC, AMR, and MLST prediction. This study offers a comprehensive overview of *S. pneumoniae* genomics and a guide for clinical and research laboratories seeking to adopt bacterial sequencing by providing important considerations when choosing sequencing platforms and analysis pipelines. We report a strong case for the implementation of WGS in the clinical setting, based on its high concordance with conventional molecular and phenotypic methods. Furthermore, the flexibility and portability of the investigated pipelines facilitate their use in clinical applications.

**Peer Reviewers** Xianwu Guo, Lab of Biotecnología Genómica, Centro de Biotecnología Genómica, Instituto de Politécnico Nacional, Tamaulipas, Mexico; Chien-Shun Chiou, Taiwan Centers for Disease Control, Taichung, Thailand

Address correspondence to Ghassan S. Dbaibo, gdbaibo@aub.edu.lb, or Antoine Abou Fayad, aa328@aub.edu.lb.

G.S. Dbaibo reports receiving research grants to his institution, serving on advisory boards, and receiving honoraria for lectures from Pfizer, MSD, Sanofi, and GSK. The authors have no other relevant affiliations or financial involvement with any organization or entity with a financial interest in or financial conflict with the subject matter or material discussed in the manuscript.

See the funding table on p. 14.

**KEYWORDS** *Streptococcus pneumoniae*, genomics, Illumina, ONT, whole-genome sequencing, hybrid assembly

*S*treptococcus pneumoniae is a highly pathogenic bacterium responsible for invasive pneumococcal diseases (IPD) such as pneumonia, meningitis, and sepsis (1). In the year 2000, S. *pneumoniae* caused 826,000 deaths globally in children aged 1–59 months (2). From 2000 to 2015, pneumococcal disease-associated deaths declined by 51% due to the introduction of pneumococcal conjugate vaccines (3). However, the battle against *S. pneumoniae* remains fierce, as non-vaccine serotypes are emerging, and resistance against multiple antimicrobial agents is developing (4–6). Over the last decade, its resistance rate against two or more antibiotic classes has increased (7). Thus, practical approaches for the rapid and precise identification of pneumococcal serotypes and resistance profiles are highly needed.

Whole-genome sequencing (WGS) is a tool that enables the high-resolution genotyping of bacterial pathogens (8). It facilitates the precise identification of bacterial strains, tracking their transmission, and informing targeted interventions (9). WGS-based approaches for pneumococcal characterization represent an attractive alternative to conventional typing methods, which can be time-consuming and labor-intensive (10, 11).

Due to its high throughput and accuracy, Illumina short-read sequencing is widely adopted for microbial genomics. However, Illumina entails high costs of sample processing, generates fragmented genomes, and lacks the ability to resolve highly repetitive genomic regions (12). Boostrom et al. estimated the start-up costs for ONT and Illumina equipment and consumables capable of generating 48 isolates (13). Based on UK prices, the cost was estimated to be £ 11,369 for ONT/Mk1c and £ 107,117 for Illumina/Miseq. By contrast, long-read sequencing from Oxford Nanopore Technologies (ONT) provides a scalable, portable, and more flexible platform for sequencing at much lower costs. Despite having lower precision, ONT results in less fragmented, more contiguous genome assemblies (14). The reduced need for large-scale sample batching, compared to Illumina, contributes to the promising potential of ONT as a diagnostic technique that can be adopted in the clinical setting.

Illumina sequencing of *S. pneumoniae* is firmly established (15, 16). However, there are limited data on bacterial genomics from ONT. The primary objective of this study was to evaluate the capacity of ONT to perform accurate predictions of serotypes and antimicrobial resistance (AMR) profiles in *S. pneumoniae* isolates compared to conventional typing techniques and Illumina. Moreover, we aimed to explore whether the newer chemistry from ONT using R10.4.1 flow cells (ONT_V14) improves the data outputs from long-read sequencing, compared to R9.4.1 flow cells and Kit10 chemistry (ONT_V10). Another objective was to compare the reads and assembly quality, the multilocus sequence typing (MLST), and lineage predictions between the different sequencing platforms and pipelines investigated.

Moreover, we assessed and compared two bioinformatics approaches for the hybrid assembly of *S. pneumoniae* genomes. Wick et al. proposed that utilizing both ONT and Illumina data to generate a hybrid assembly is the optimal approach for obtaining a perfect bacterial genome (17). Hybrid assembly can be performed by building a short-read assembly graph first, then using long reads for scaffolding (18). Another approach is long-read assembly followed by short-read polishing (19).

## MATERIALS AND METHODS

### Study design

The study included a total of 27 *S. pneumoniae* bacteria that were collected as part of the Lebanese Inter-Hospital Pneumococcal Surveillance Program (LIPSP). The latter is an ongoing national surveillance program established in 2005 to track the changing landscape of invasive pneumococcal disease in Lebanon. It was approved by the AUB

Institutional Review Board (IRB) (PED.GD.01). *S. pneumoniae* were isolated from the cerebrospinal fluid (CSF) or blood of patients hospitalized with IPD between February 2017 and July 2023 and stored at −80°C until further processing. Around half of the patients (48%) belonged to the ≤5 years age group (Table S1). The study design is summarized in Fig. S1.

## DNA extraction

For sequencing, *S. pneumoniae* isolates were cultured from frozen stocks on sheep blood agar plates and incubated overnight in 5% $CO_2$ at 37°C. They were then sub-cultured in 5 mL of Todd Hewitt enrichment broth supplemented with 0.5% yeast extract and rabbit serum. After centrifugation at 5,000 × *g* for 5 minutes, double-stranded DNA was extracted from the bacterial pellet and ~180 µL supernatant using the Quick-DNA HMW MagBead Kit (Zymo Research, USA). The latter is a magnetic bead-based system for high molecular weight genomic DNA extraction. DNA was eluted in 40 µL elution buffer and quantified using the Qubit fluorometer (Life Technologies, Carlsbad, California, US) with the High Sensitivity dsDNA Assay Kit. Purity was assessed using the NanoDrop Spectrophotometer. The same DNA extract was used for both long‑read ONT and short‑read Illumina sequencing, where possible. For serotyping, DNA was extracted before storage as part of routine surveillance using the InstaGene matrix (BioRad, USA). All kits were used according to the manufacturer's instructions.

## Conventional microbial typing

A sequential multiplex PCR technique composed of seven reactions with different primer sets was utilized for serotyping as previously described (6). Each reaction included a negative control and an internal positive control for a conserved region in the pneumococcal *cps* operon. If the serotype was not detected by all seven reactions, an isolate was classified as non-typeable (NT). Antimicrobial susceptibility testing (AST) was performed using disk diffusion to determine resistance profiles against chloramphenicol, clindamycin, erythromycin, sulfamethoxazole/trimethoprim, tetracycline, and levofloxacin and the E-test (Biomerieux, France) for penicillin following the CLSI guidelines (20). Penicillin minimum inhibitory concentration (MIC) was interpreted based on the meningitis MIC cut-off. Penicillin with MIC >0.06 µg/mL was categorized as resistant.

## Library preparation and sequencing

In the first sequencing round, a total of 31 *S. pneumoniae* bacteria were sequenced using the ONT Mk1c system on R9.4.1 flow cells with Kit10 chemistry (ONT_V10). DNA libraries were prepared from 50 ng of genomic DNA per sample using the Rapid Barcoding Kit 96 (SQK-RBK110.96) following the manufacturer's protocol. The total combined genome size of the bacterial samples loaded into each R9.4.1 flow cell (FLO-MIN106) was limited to 35 million base pairs. All ONT sequencing was performed on the MinION Mk1C device (Oxford Nanopore Technologies, UK) for 72 hours. Guppy, a base caller integrated within the MinKNOW software, was utilized for the live base calling of ONT_V10 reads using the FAST base calling model.

In the second sequencing round, isolates that passed taxonomy quality control (QC) in the first round (*n* = 27) were sequenced using the Illumina platform. Short‑read Illumina libraries were prepared using the Illumina DNA prep kit (Illumina, San Diego, CA) per the manufacturer's protocol. The DNA library was prepared from 500 ng of total genomic DNA per sample. The MiSeq V2 Reagent Kit was used for sequencing the pooled DNA library on the Illumina MiSeq platform (Illumina, San Diego, CA) to generate paired‑end, 250 bp indexed reads. Illumina sequencing required 72 hours to be completed.

Furthermore, a total of 12 isolates were selected for sequencing using the most recent kit 14 chemistry and R10.4.1 flow cells from ONT (ONT_V14). The rationale for selecting these isolates is detailed in Fig. S1. The Rapid Barcoding Kit 96 V14 (SQK-RBK114.96) was

utilized for library preparation from 50 ng of genomic DNA per sample. Live base calling of ONT_V14 reads was performed using Dorado (v7.1.4) and the FAST base calling model.

## Data analysis

The fastq-scan tool was used to generate raw data statistics for both MiSeq and MinION data. Illumina data were analyzed using the GPS pipeline (21), which utilizes SPAdes (v1.1.0) for assembly and QUAST (v5.0.2) for assembly QC.

FastQ files from long-read sequencing were analyzed using the assemble-BAC-ONT pipeline (22) version 1.1 to generate polished FASTA files using Flye (v2.9-b1768) and medaka (v1.4.4). The pipeline additionally generates QC reports using MultiQC and checkm2 (v1.0.1).

FASTA files obtained from both pipelines were analyzed using Pathogenwatch (https://pathogen.watch/) to obtain serotyping, AMR, MLST, and Global Pneumococcal Sequence Clusters (GPSC) predictions. Pathogenwatch utilizes SeroBA, a *k*-mer-based method, to predict the serotypes of *S. pneumoniae*. It employs the PAARSNP software to infer the presence or absence of resistance genetic markers by using BLAST against a custom AMR database. Moreover, it performs penicillin-binding protein (pbp) typing based on the method published by Li et al. (23). The predicted resistance profiles are then interpreted using the CLSI guidelines. Pathogenwatch predicts sequence types (STs) using the MLST scheme hosted in the PubMLST database.

The serotypes of all isolates were additionally predicted using PneumoKITy (24), which is a sensitive *k*-mer-based tool that allows the prediction of mixed *S. pneumoniae* serotypes. PneumoKITY was used to predict the serotypes of all isolates sequenced with Illumina, ONT_V10, and ONT_V14.

ONT FastQ files generated using kit10 chemistry were additionally analyzed with Porechop (v0.2.4) (https://github.com/rrwick/Porechop) for trimming and Canu (v2.2) (25) for assembly. The outputs from Canu were compared to those from the assemble-BAC-ONT pipeline for the same isolates to determine the better approach for assembling long-read data.

Hybrid assembly was performed using data from ONT with kit10 chemistry and Illumina. Two approaches for generating hybrid assemblies were compared. The first approach relies on short-read-first assembly with Unicycler (v0.5.0) (18), which uses long reads to scaffold a short-read assembly graph to completion. The input for Unicycler is Illumina and ONT FastQ files. The second approach utilizes Pilon (v1.24) (26) and relies on polishing ONT assemblies with short reads. For phylogenetic analysis, annotations from hybrid genomes (*n* = 27) were generated using bakta (v1.8.2) (27). A core genome alignment of the *S. pneumoniae* sequences and a reference genome (GPSC46) from the global pneumococcal sequencing project was generated using Panaroo (v1.3.4) (28). The snp-sites (v2.5.1) tool (29) was utilized to extract variable sites and determine the number of constant sites in the initial alignment. The tree was inferred with iqtree (v2.2.3) (30) using the maximum likelihood method and visualized with Microreact (https://micro-react.org/).

GraphPad for Windows (v8.4) (GraphPad Software, California, USA) was used for statistical analysis. Normality was assessed using the Kolmogorov–Smirnov test, and mean quality scores were compared using the Mann-Whitney test.

## RESULTS

### Quality of raw data

The fastq-scan tool was used for raw data quality assessment. The median coverage was 58.8× and 77.29× for ONT_V10 and Illumina sequences, respectively (Fig. 1A). MiSeq sequencing generated substantially more reads than ONT_V10 (Fig. 1B). However, the average read length in ONT_V10 was higher than that of Illumina (Fig. 1C). The maximum read length was impressively high in ONT_V10, ranging from 17,469 bp to 288,361 bp (Table S2).

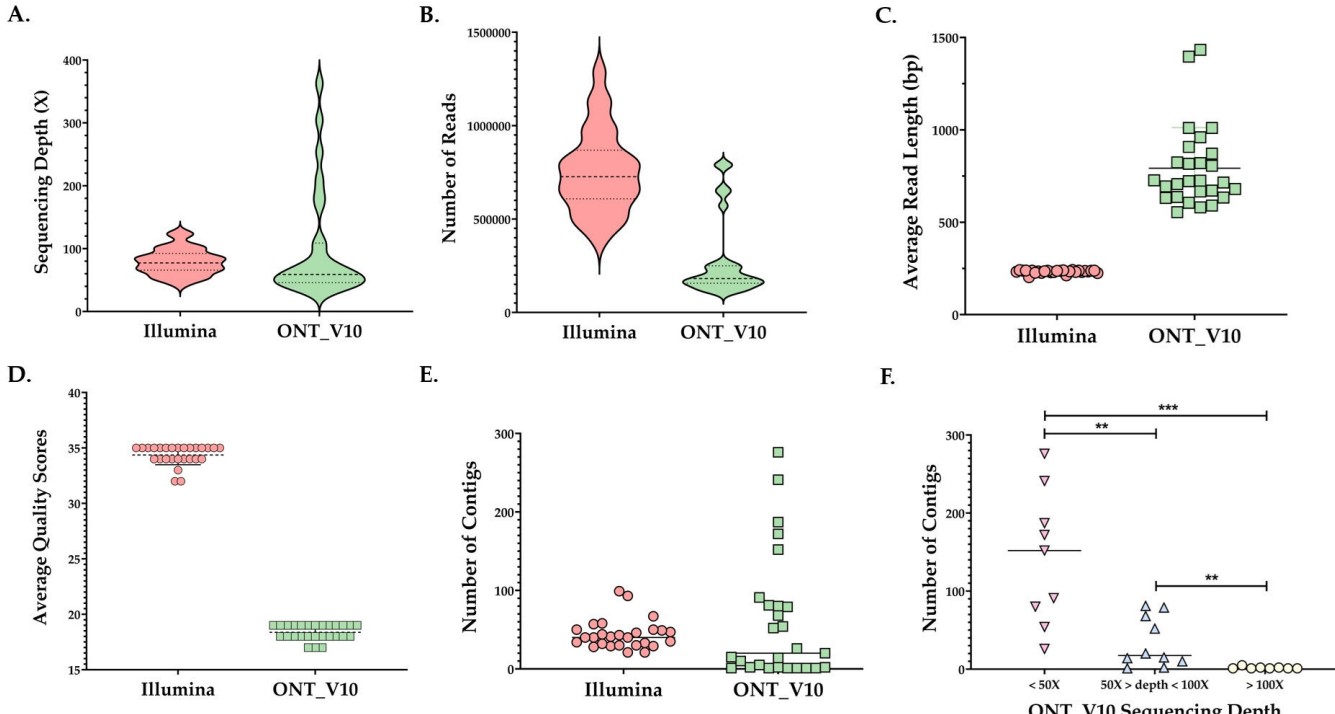

**FIG 1** Quality of raw data and assembled genomes obtained from *Streptococcus pneumoniae* isolates (*n* = 27) sequenced by both Illumina and ONT_V10 platforms. Raw data were compared between both techniques according to the median sequencing depth (A), number of reads (B), and the averages of read length (C) and quality scores (D). Panel E represents the median number of contigs in assemblies generated from Illumina or ONT data. Panel F represents the number of contigs obtained from ONT sequencing according to sequencing depth cut-offs of 50× and 100×. Panel G plots the variation in contiguity according to ONT sequencing depth. Significant values are indicated with **\*\*P* < 0.01 or \*\*\**P* < 0.001 (Mann-Whitney test). Abbreviations: bp, Base pairs; ONT_V10, Oxford Nanopore Technologies (R9.4.1 flow cells/ Kit10 chemistry).

Figure 1D shows that the average quality score of Illumina sequences (*n* = 27; mean, 34.37; SD, 0.88) was higher than that of ONT_V10 (*n* = 27; mean, 18.37; SD, 0.68). These quality scores correspond to Phred scores, which quantify the error probability in a base call.

## Quality of assembled data

ONT_V10 genome assemblies were compared between porechop/Canu and the assemble-BAC-ONT pipeline. The latter enhanced contiguity and GPSC identification compared to Canu (Table S3). Moreover, the pipeline generated assemblies with reduced genome size in 24/26 isolates. Canu took a long time—up to 6 hours in one isolate—to finalize the *de novo* assembly. The assemble-Bac-ONT pipeline was configured to utilize 11 .GB max_memory, 6 max_cpus, and 12 .h max_time to overcome local computational limitations. Despite this, it was generally fast, taking around 3 hours to complete the analysis of 11 samples. We utilized the assemble-BAC-ONT pipeline as the standard approach for ONT analysis.

The median contig number was 20 in ONT_V10 (IQR = 79) and 40 in Illumina (IQR = 20) (Fig. 1E). The number of contigs ranged between 1 and 5 in ONT_V10 samples with sequencing depth >100× (Fig. 1F). Isolates with depth >100× had significantly lower contigs compared to those with depths of 50× to 100× (*P* = 0.002) and to isolates with depth <50× (*P* < 0.0001). Furthermore, all isolates sequenced by Illumina passed genome size QC compared to 25/27 (93 %) in ONT_V10.

## Serotyping and global pneumococcal sequence clusters

Sample S664 (1/27) was NT by PCR and, therefore, not included in the concordance analysis between PCR and WGS. The serogroups of 23/26 (88 %) isolates predicted by WGS were concordant with multiplex PCR (Table 1). Moreover, all WGS-based serogroups were concordant between Illumina and ONT_V10 (100%). In all isolates that were assigned to serogroup 24 by Pathogenwatch ($n = 3$), PneumoKity consistently identified serotype 24F2 as the one with the highest percentage identity. Moreover, discrepancies between Pathogenwatch and PneumoKITY were observed in isolate S524 at the serotype but not the serogroup level. GPSC were identified in all isolates by Illumina and in 24/27 isolates by ONT_V10. Among isolates with GPSC assigned by both ONT and Illumina ($n = 24$), the concordance in predicted GPSC was 100%.

## Antimicrobial resistance

The concordance in AMR results between the different sequencing platforms was investigated. AMR profiles against chloramphenicol, clindamycin, erythromycin, and tetracycline were concordant between Illumina and ONT_V10 in all isolates (100%) (Table 2). Trimethoprim and sulfamethoxazole resistance was concordant between ONT_V10 and Illumina in 26/27 (96 %) isolates. In the discordant sample S647, resistance to sulfamethoxazole/trimethoprim was confirmed by AST and Illumina. Repeating S647 by ONT_V14 revealed resistance markers against sulfamethoxazole (*folP_aa_insert_57–70*).

The concordance in AMR results was also investigated for genotypic approaches in comparison to AST. AMR results for chloramphenicol and erythromycin were 100% concordant between WGS and AST. Moreover, the concordance between both approaches against sulfamethoxazole/trimethoprim was 93% (25/27). Both Illumina and ONT_V10 revealed resistance markers against tetracycline in three samples (11%) that were susceptible by AST, namely *tetM_12* in S647, *tetM_4* in S673, and *tetM_8* in S674. The concordance in fluoroquinolone resistance between AST and Illumina or ONT_V14 was 93% (25/27). S678 and S686 were predicted to be sensitive to fluoroquinolones by all WGS techniques, but intermediate or resistant by AST. Interestingly, ONT_V10 predicted the presence of resistance markers against fluoroquinolones in S645 and S647, which were determined susceptible by AST. ONT_V14 revealed the lack of fluoroquinolone resistance markers in both isolates. Furthermore, the concordance between Illumina and ONT_V10 was 100% for ceftriaxone, kanamycin, and linezolid (data not shown).

Penicillin resistance and MICs were determined by Illumina in all isolates and 21/27 (78 %) isolates by ONT_V10 (Table S5). ONT_V14 data were available for three isolates that lacked penicillin resistance and MIC data in ONT_V10 (Table S6). In samples S645 and S678, ONT_V14 generated the exact data for resistance, MIC and pbp types as Illumina. In S664, the pbp1a and pbp2a types predicted by ONT_V14 matched those from Illumina. ONT_V14 data were also available for samples with discordant pbp data between ONT_V10 and Illumina ($n = 6$). ONT_V14 generated pbp and MIC data that completely matched those from Illumina in three samples (S653, S668, and S683).

## Multi-locus sequence typing alleles and sequence types

To maximize uniformity across results, MLST obtained from Pathogenwatch for all sequencing approaches were analyzed. All MLST alleles were identified by Illumina in all isolates (100%) (Table 3). Novel STs were identified by Illumina in three isolates, *b03f in S670, *fec1 in S683, and *e47f in S686. All serogroup 24 isolates were identified as ST 15253.

With ONT_V10, 24/27 (89 %) isolates were assigned novel STs due to missing data in one or more MLST alleles. The ST of the remaining isolates (S524, S649, and S674) was concordant with those obtained in Illumina. Interestingly, these isolates did not have the highest ONT_V10 sequencing depth. The ONT_V10 depth in S650 and S644, in which only four MLST alleles were identified, was 362× and 211×, respectively.

**TABLE 1** GPSCs and serotyping results of *S. pneumoniae* isolates (*n* = 27) assessed by multiplex PCR and WGS[b]

| Sample | Multiplex PCR | Serotyping Pathogenwatch[a] Illumina | ONT_V10 | ONT_V14 | PneumoKITy top hit Illumina (Max kmer percentage) | ONT_V10 (Max kmer percentage) | ONT_V14 (Max kmer percentage) | GPSC strain Illumina | ONT_V10 | ONT_V14 |
|---|---|---|---|---|---|---|---|---|---|---|
| S515 | 22F/22A | 22F | 22F | | 22F (99.1%) | 22F (97.5%) | | 19 | 19 | |
| S516 | 3 | 3 | 3 | | 3 (95.2%) | 3 (93.8%) | | 12 | 12 | |
| S524 | 9V/9A | 9V | 9V | | 9A (95.6%) | 9A (94.2%) | | 61 | 61 | |
| S531 | 1 | 1 | 1 | | 1 (98.2%) | 1 (98.1%) | | 31 | 31 | |
| S644 | 8 | 8 | 8 | | 8 (99.8%) | 8 (99.2%) | | 3 | 3 | |
| S645 | 23A | 23A | 23A | 23A | 23A (99.6%) | 23A (97.1%) | 23A (98.9%) | 7 | 7 | 7 |
| S647 | 18 | 18C | 18C | 18C | 18C (91.3%) | 18C (89.9%) | 18C (90.3%) | 67 | 67 | 67 |
| S649 | 3 | 38 | 38 | | 38 (89.7%) | 38 (89%) | | 38 | 38 | |
| S650 | 8 | 8 | 8 | 8 | 8 (99.8%) | 8 (99.1%) | 8 (99.1%) | 3 | 3 | 3 |
| S651 | 3 | 3 | 3 | | 3 (95.2%) | 3 (94.5%) | | 12 | 12 | |
| S653 | 4 | 17F | 17F | 17F | 17F (93.5%) | 17F (87.4%) | 17F (93.5%) | 10 | 10 | 10 |
| S660 | 14 | 14 | 14 | | 14 (92.9%) | 14 (92.1%) | | 6 | 6 | |
| S664 | NT | Serogroup 24 | Serogroup 24 | Serogroup 24 | 24F2 (99.7%) | 24F2 (98.9%) | 24F2 (99.7%) | 10 | 10 | 10 |
| S666 | 33F | 33F | 33F | | 33F (100%) | 33F (97.7%) | | 3 | 3 | 3 |
| S668 | 5 | 5 | 5 | 5 | 5 (95.3%) | 5 (94.7%) | 5 (94.9%) | 8 | 8 | 8 |
| S669 | 15A | 15A | 15A | 15A | 15A (99.8%) | 15A (94.7%) | 15A (99.8%) | 8 | Not assigned | Not assigned |
| S670 | 12F/12A | 15C | 15C | 15C | 15C (87.9%) | 15C (87.2%) | 15C (87.7%) | 5 | 5 | 5 |
| S673 | 33F | 33F | 33F | 33F | 33F (100%) | 33F (97.7%) | 33F (99.8%) | 3 | 3 | 3 |
| S674 | 24A/24B/24F | Serogroup 24 | Serogroup 24 | | 24F2 (99.6%) | 24F2 (98.2%) | | 10 | 10 | |
| S676 | 24A/24B/24F | Serogroup 24 | Serogroup 24 | | 24F2 (99.5%) | 24F2 (97.9%) | | 10 | 10 | |
| S678 | 9N/9L | 9N | 9N | 9N | 9N (100%) | 9N (99.3%) | 9N (99.8%) | 699 | 699 | 699 |
| S682 | 15B/C | 15B | 15C | | 15C (95.3%) | 15C (93.6%)/ 15B (93.6%) | | 10 | 10 | |
| S683 | 15B/C | 15B | 15C | 15B | 15C (95.8%) | 15C (94.7%)/ 15B (94.7%) | 15C (95.8%) | 10 | Not assigned | Not assigned |
| S685 | 19A | 19A | 19A | 19A | 19A (98.7%) | 19A (98.2%) | 19A (98.7%) | 44 | Not assigned | 44 |
| S686 | 15B/C | 15B | 15C | | 15C (87.8%) | 15C (85.9%) | | 10 | 10 | |
| S688 | 10A | 10A | 10A | | 10A (99.3%) | 10A (98.3%) | | 35 | 35 | |
| S689 | 38 | 38 | 38 | | 38 (90.2%) | 38 (88.5%) | | 38 | 38 | |

[a]Pathogenwatch is a platform with a user-friendly interface that utilizes SeroBA, a *k*-mer-based method, to predict the serotypes of *S. pneumoniae* bacteria.

[b]Abbreviations: GPSC, global pneumococcal sequence clusters; NT, not typeable; ONT_V10, Oxford Nanopore Technologies (R9.4.1 flow cells/ Kit10 chemistry); ONT_V14, Oxford Nanopore Technologies (R10.4.1 flow cells/ Kit14 chemistry); PCR, polymerase chain reaction.

TABLE 2  Predicted resistance profiles by the different sequencing technologies compared to AST results in *S. pneumoniae* isolates (*n* = 27) causing IPD[a]

| Isolate | Chloramphenicol | | | | Clindamycin | | | | Erythromycin | | | | Tetracycline | | | | Trimethoprim/ sulfamethoxazole | | | | Fluoroquinolones | | | |
|---|---|---|---|---|---|---|---|---|---|---|---|---|---|---|---|---|---|---|---|---|---|---|---|---|
| | AST | ILL | ONT V10 | ONT V14 | AST | ILL | ONT V10 | ONT V14 | AST | ILL | ONT V10 | ONT V14 | AST | ILL | ONT V10 | ONT V14 | AST | ILL | ONT V10 | ONT V14 | AST | ILL | ONT V10 | ONT V14 |
| S515 | S | S | S | – | S | S | S | – | S | S | S | – | S | S | S | – | S | S | S | – | S | S | S | – |
| S516 | S | S | S | – | S | S | S | – | S | S | S | – | S | S | S | – | S | S | S | – | S | S | S | – |
| S524 | S | S | S | – | S | S | S | – | S | S | S | – | R | R | R | – | S | R | R | – | S | S | S | – |
| S531 | S | S | S | – | S | S | S | – | S | S | S | – | S | S | S | – | S | S | S | – | S | S | S | – |
| S644 | S | S | S | – | S | S | S | – | S | S | S | – | S | S | S | – | S | S | S | – | S | S | S | – |
| S645 | S | S | S | S | S | S | S | S | S | S | S | S | S | S | S | S | S | S | S | S | S | S | R | S |
| S647 | S | S | S | S | R | S | S | S | R | R | R | R | S | R | R | R | R | R | S | R | S | S | R | S |
| S649 | S | S | S | – | S | S | S | – | R | R | R | – | R | R | R | – | R | R | R | – | S | S | S | – |
| S650 | S | S | S | S | S | S | S | S | S | S | S | S | S | S | S | S | S | S | S | S | S | S | S | S |
| S651 | S | S | S | – | S | S | S | – | S | S | S | – | S | S | S | – | S | S | S | – | S | S | S | – |
| S653 | S | S | S | S | R | R | R | R | R | R | R | R | R | R | R | S | R | S | S | S | S | S | S | S |
| S660 | S | S | S | – | R | R | R | R | R | R | R | R | R | R | R | – | I | R | R | – | S | S | S | – |
| S664 | S | S | S | S | R | R | R | R | R | R | R | R | I | R | R | R | R | R | R | R | S | S | S | S |
| S666 | S | S | S | – | S | S | S | – | S | S | S | – | S | S | S | – | S | S | S | – | S | S | S | – |
| S668 | S | S | S | S | S | S | S | S | S | S | S | S | S | S | S | S | R | R | R | R | S | S | S | S |
| S669 | S | S | S | S | S | S | S | S | S | S | S | S | S | S | S | S | S | S | S | S | S | S | S | S |
| S670 | S | S | S | S | S | S | S | S | S | S | S | S | S | S | S | S | S | S | S | S | S | S | S | S |
| S673 | S | S | S | S | R | R | R | R | R | R | R | R | S | R | R | R | R | S | S | S | S | S | S | S |
| S674 | S | S | S | – | R | R | R | – | R | R | R | – | S | R | R | – | R | R | R | – | S | S | S | – |
| S676 | S | S | S | – | R | R | R | – | R | R | R | – | I | R | R | – | R | R | R | – | S | S | S | – |
| S678 | S | S | S | S | R | R | R | R | R | R | R | R | R | R | R | R | R | R | R | R | S | S | S | S |
| S682 | S | S | S | – | R | R | R | – | R | R | R | – | R | R | R | – | R | R | R | – | S | S | S | – |
| S683 | S | S | S | S | R | R | R | R | R | R | R | R | R | R | R | S | R | R | R | R | S | S | S | S |
| S685 | S | S | S | S | R | R | R | R | R | R | R | R | R | R | R | R | S | S | S | S | S | S | S | S |
| S686 | S | S | S | – | R | R | R | – | R | R | R | – | R | R | R | – | R | R | R | – | R | S | S | – |
| S688 | S | S | S | – | S | S | S | – | S | S | S | – | S | S | S | – | S | S | S | – | S | S | S | – |
| S689 | S | S | S | – | S | S | S | – | R | R | R | – | R | R | R | – | S | S | S | – | S | S | S | – |

[a]Pathogenwatch was used to predict antimicrobial resistance profiles in WGS-based data. Pathogenwatch utilizes the PAARSNP software to infer the presence or absence of resistance genetic markers by using BLAST against a custom AMR database. AST, Antimicrobial susceptibility testing; ONT V10, Oxford Nanopore Technologies (R9.4.1 flow cells/ Kit10 chemistry); ONT V14, Oxford Nanopore Technologies (R10.4.1 flow cells/ Kit14 chemistry): ILL, Illumina; I, Intermediate; R, Resistant; S, Sensitive. The cells denoted by the "–" sign indicate that the isolate was not processed by ONT_V14.

TABLE 3 Comparative analysis of MLST loci between Illumina, ONT_V10, and ONT_V14 in *S. pneumoniae* isolates (*n* = 27) causing IPD[a]

| Sample | MLST alleles (*n* = 7) | | | | | | | Sequence type (ST) | Number of assigned alleles |
|---|---|---|---|---|---|---|---|---|---|
| | aroE | gdh | gki | recP | spi | xpt | ddl | | |
| S515_ONT_V10 | 1 | 1 | 4 | 1 | 18 | (novel) | 17 | *705d | 6 |
| S515_Illumina | 1 | 1 | 4 | 1 | 18 | 58 | 17 | 433 | All |
| S516_ONT_V10 | (novel) | 15 | (novel) | 10 | (novel) | 1 | 22 | *12a3 | 4 |
| S516_Illumina | 7 | 15 | 2 | 10 | 6 | 1 | 22 | 180 | All |
| S524_ONT_V10 | 1 | 16 | 4 | 10 | 7 | 22 | 5 | 16225 | All |
| S524_Illumina | 1 | 16 | 4 | 10 | 7 | 22 | 5 | 16225 | All |
| S531_ONT_V10 | (novel) | (novel) | (novel) | 5 | (novel) | (novel) | 20 | *1A05 | 2 |
| S531_Illumina | 12 | 8 | 13 | 5 | 16 | 4 | 20 | 306 | All |
| S644_ONT_V10 | 2 | 5 | (novel) | 11 | (novel) | (novel) | 14 | *f014 | 4 |
| S644_Illumina | 2 | 5 | 1 | 11 | 16 | 3 | 14 | 53 | All |
| S645_ONT_V10 | ? | ? | 9 | 9 | (novel) | (novel) | (novel) | *a568 | 2 |
| S645_ONT_V14 | (novel) | 8 | 9 | 9 | ? | (novel) | (novel) | *13b0 | **3** |
| S645_Illumina | 1 | 8 | 9 | 9 | 6 | 4 | 6 | 42 | All |
| S647_ONT_V10 | (novel) | 11 | 34 | 16 | (novel) | (novel) | (novel) | *5881 | 3 |
| S647_ONT_V14 | 10 | 11 | 34 | 16 | 15 | 1 | (novel) | *b33c | 6 |
| S647_Illumina | 10 | 11 | 34 | 16 | 15 | 1 | 145 | 1233 | All |
| S649_ONT_V10 | 1 | 43 | 41 | 18 | 13 | 49 | 8 | 9325 | All |
| S649_Illumina | 1 | 43 | 41 | 18 | 13 | 49 | 8 | 9325 | All |
| S650_ONT_V10 | 2 | 5 | (novel) | 18 | (novel) | (novel) | 18 | *b965 | 4 |
| S650_ONT_V14 | 2 | 5 | 29 | 18 | 42 | (novel) | 18 | *e476 | 6 |
| S650_Illumina | 2 | 5 | 29 | 18 | 42 | 3 | 18 | 1012 | All |
| S651_ONT_V10 | 7 | 15 | 2 | 10 | (novel) | (novel) | 22 | *6e8f | 5 |
| S651_Illumina | 7 | 15 | 2 | 10 | 6 | 1 | 22 | 180 | All |
| S653_ONT_V10 | ? | 19 | 2 | 17 | (novel) | 22 | 17 | *1d15 | 5 |
| S653_ONT_V14 | 2 | 19 | 2 | ? | 15 | 22 | 17 | *b093 | 6 |
| S653_Illumina | 2 | 19 | 2 | 17 | 15 | 22 | 17 | 2355 | All |
| S660_ONT_V10 | 7 | 5 | (novel) | 18 | (novel) | (novel) | 1 | *d6fb | 4 |
| S660_Illumina | 7 | 5 | 10 | 18 | 6 | 8 | 1 | 143 | All |
| S664_ONT_V10 | ? | (novel) | 2 | 17 | (novel) | (novel) | 14 | *8023 | 3 |
| S664_ONT_V14 | 5 | 19 | 2 | 17 | 10 | 22 | 14 | 15253 | All |
| S664_Illumina | 5 | 19 | 2 | 17 | 10 | 22 | 14 | 15253 | All |
| S666_ONT_V10 | 8 | 5 | (novel) | 66 | (novel) | 3 | 5 | *9622 | 5 |
| S666_Illumina | 8 | 5 | 29 | 66 | 42 | 3 | 5 | 2223 | All |
| S668_ONT_V10 | 16 | 12 | 9 | 1 | (novel) | 33 | 33 | *0038 | 6 |
| S668_ONT_V14 | 16 | 12 | 9 | 1 | 41 | 33 | 33 | 289 | All |
| S668_Illumina | 16 | 12 | 9 | 1 | 41 | 33 | 33 | 289 | All |
| S669_ONT_V10 | 10 | 16 | 34 | 16 | (novel) | (novel) | 31 | *8b24 | 5 |
| S669_ONT_V14 | 10 | 16 | 34 | 16 | 6 | ? | 31 | *9b15 | 6 |
| S669_Illumina | 10 | 16 | 34 | 16 | 6 | 1 | 31 | 3376 | All |
| S670_ONT_V10 | 7 | 13 | 8 | 42 | (novel) | (novel) | 8 | *60af | 5 |
| S670_ONT_V14 | 7 | 13 | 8 | 42 | 6 | (novel) | **8** | *60af | 6 |
| S670_Illumina | 7 | 13 | 8 | 42 | 6 | 6 | 8 | *b03f | All |
| S673_ONT_V10 | 5 | 35 | (novel) | 1 | (novel) | 39 | 18 | *9ece | 5 |
| S673_ONT_V14 | 5 | 35 | 29 | 1 | 45 | 39 | 18 | 717 | All |
| S673_Illumina | 5 | 35 | 29 | 1 | 45 | 39 | 18 | 717 | All |
| S674_ONT_V10 | 5 | 19 | 2 | 17 | 10 | 22 | 14 | 15253 | All |
| S674_Illumina | 5 | 19 | 2 | 17 | 10 | 22 | 14 | 15253 | All |
| S676_ONT_V10 | 5 | (novel) | 2 | 17 | (novel) | 22 | 14 | *b8a0 | 5 |
| S676_Illumina | 5 | 19 | 2 | 17 | 10 | 22 | 14 | 15253 | All |
| S678_ONT_V10 | 2 | ? | (novel) | 1 | 7 | 1 | 5 | *e123 | 5 |
| S678_ONT_V14 | 2 | ? | 174 | 1 | 7 | 1 | 5 | *ea64 | 6 |

(*Continued on next page*)

**TABLE 3** Comparative analysis of MLST loci between Illumina, ONT_V10, and ONT_V14 in *S. pneumoniae* isolates (*n* = 27) causing IPD[a] (*Continued*)

| Sample | MLST alleles (*n* = 7) | | | | | | | Sequence type (ST) | Number of assigned alleles |
|--------|------|------|------|------|------|------|------|------|------|
| | aroE | gdh | gki | recP | spi | xpt | ddl | | |
| S678_Illumina | 2 | 5 | 174 | 1 | 7 | 1 | 5 | 6359 | All |
| S682_ONT_V10 | 12 | 19 | 2 | 1 | (novel) | (novel) | 26 | *2214 | 5 |
| S682_Illumina | 12 | 19 | 2 | 1 | 6 | 22 | 26 | 13727 | All |
| S683_ONT_V10 | 12 | (novel) | 2 | 9 | 687 | 22 | 9 | *ab36 | 6 |
| S683_ONT_V14 | 12 | 19 | 2 | 9 | 687 | 22 | 9 | *fec1 | All |
| S683_Illumina | 12 | 19 | 2 | 9 | 687 | 22 | 9 | *fec1 | All |
| S685_ONT_V10 | 7 | 14 | 40 | (novel) | 1 | 1 | 14 | *7c13 | 6 |
| S685_ONT_V14 | 7 | 14 | 40 | 12 | 1 | (novel) | 14 | *8991 | 6 |
| S685_Illumina | 7 | 14 | 40 | 12 | 1 | 1 | 14 | 179 | All |
| S686_ONT_V10 | 12 | (novel) | 2 | 6 | (novel) | 22 | 9 | *42a5 | 5 |
| S686_Illumina | 12 | 19 | 2 | 6 | 6 | 22 | 9 | *e47f | All |
| S688_ONT_V10 | 5 | (novel) | 4 | 2 | (novel) | 1 | 6 | *f200 | 5 |
| S688_Illumina | 5 | 7 | 4 | 2 | 10 | 1 | 6 | 1551 | All |
| S689_ONT_V10 | 1 | 43 | 41 | 18 | (novel) | 49 | 8 | *b0d8 | 6 |
| S689_Illumina | 1 | 43 | 41 | 18 | 13 | 49 | 8 | 9325 | All |

[a]MLST, Multilocus sequence typing; aroE, Shikimate dehydrogenase; gdh, glucose-6-phosphate dehydrogenase; gki, glucose kinase; recP, transketolase; spi, signal peptidase I; xpt, xanthine phosphoribosyltransferase; ddl, D-Alanine-D-alanine ligase; ONT_V10, Oxford Nanopore Technologies (R9.4.1 flow cells/ Kit10 chemistry); ONT_V14, Oxford Nanopore Technologies (R10.4.1 flow cells/ Kit14 chemistry.

We aimed to explore whether MLST results were improved in the 12 isolates sequenced using ONT_V14. All MLST alleles were identified in 4/12 (33%) isolates processed with ONT_V14 (Table 3). In these four isolates (S664, S668, S673, and S683), the ST matched those from Illumina. In 11/12 (92 %) isolates, the number of alleles identified by ONT_V14 was higher than that determined by ONT_V10 chemistry. In S650, ONT_V14 at a sequencing depth of 55× increased the number of identified alleles from 4/7 to 6/7 compared to ONT_V10 (362×).

## Hybrid assembly and phylogeny

The genomes of all isolates were reconstructed by combining data from ONT_V10 and Illumina. Hybrid assembly was performed using two different approaches (Fig. 2). The first was short-read-first hybrid assembly using Unicycler. The second included long-read assembly followed by short-read polishing using Pilon. Notably, Pilon was less time-consuming, taking around 4 min/sample for hybrid assembly. Unicycler required ~60 min to process files with ONT depth >100×.

For samples with ONT depth >100× (*n* = 8), the Pilon approach generated genomes with improved QC parameters in 4/8 isolates compared to Unicycler (Table S7). In the remaining four isolates, the QC parameters were almost identical to Unicycler. At ONT depths <50× (*n* = 9), Unicycler generated higher N50 parameters and fewer contigs compared to Pilon (Table S8). At ONT depths between 50× and 76× (*n* = 10), Unicycler improved the N50 and contig number parameters in 7/10 isolates (70%).

The predicted serotypes, GPSC and AMR markers from Pilon, and Unicycler were identical in all isolates (*n* = 27) (data not shown). Unicycler assembly graphs visualized by the Bandage software revealed that most isolates were assembled into circular structures (Fig. S2).

Phylogenetic analysis showed that isolates with simultaneous predicted resistance against penicillin, clindamycin, erythromycin, tetracycline, and sulfamethoxazole clustered together (Fig. 3). All serogroup 24 and serotype 15B isolates in this study belonged to these clusters.

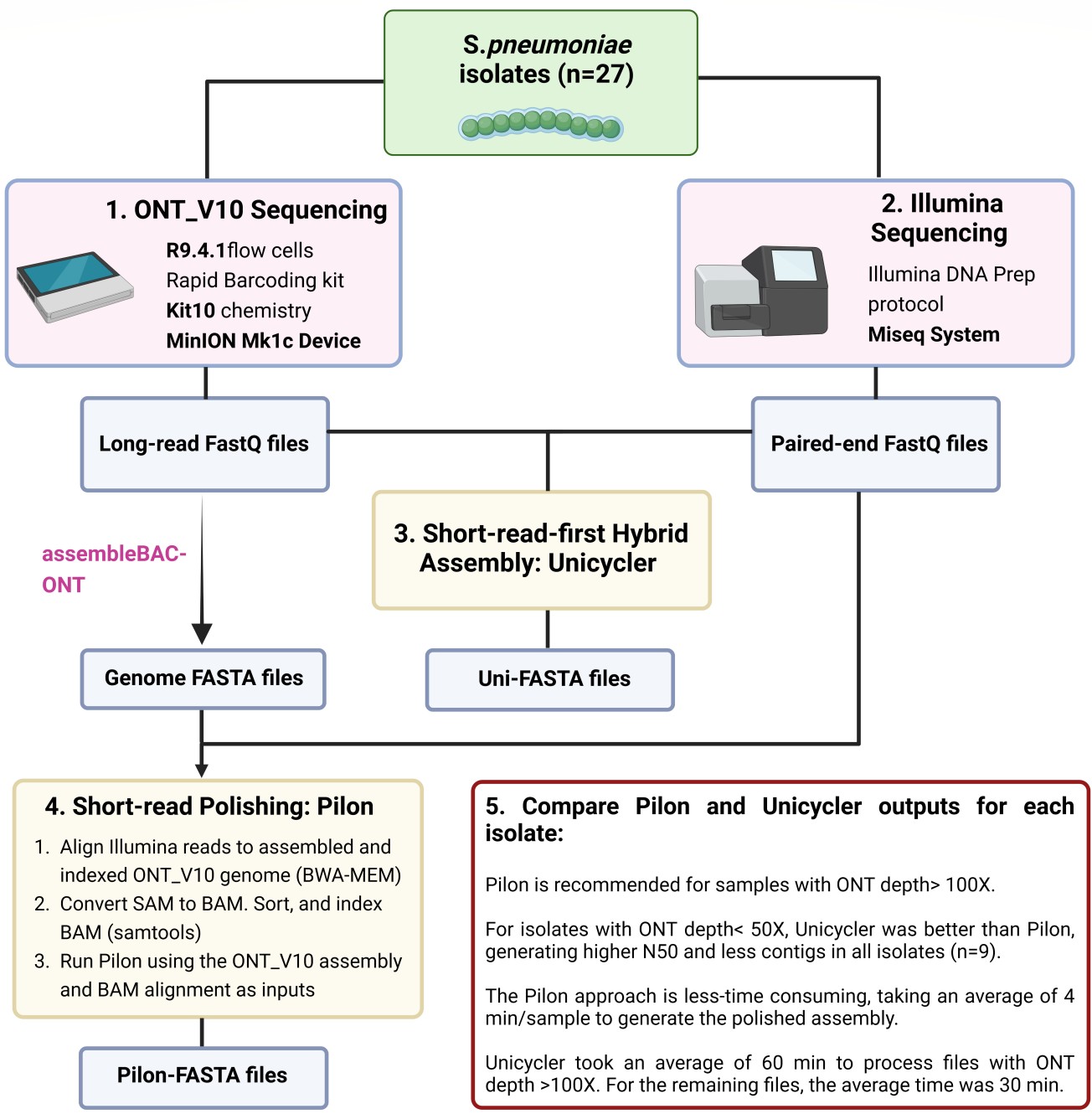

**FIG 2** Illustrated overview of the workflow utilized to compare hybrid assemblies generated by Unicycler and Pilon for S. *pneumoniae isolates* (n = 27). *S. pneumoniae* bacteria were sequenced using the ONT Mk1c system with R9.4.1 flowcells and Kit10 chemistry (1) and the Illumina Miseq system (2). Illumina and ONT_V10 FastQ files were processed using Unicycler, which utilizes long reads to scaffold a short-read assembly graph to completion (3). ONT_V10 FASTA files were indexed and used to align Illumina reads (4). The resulting SAM files were processed by samtools and utilized by Pilon for hybrid assembly. The genome quality results from both hybrid assembly approaches were compared (5). Abbreviations: ONT_V10, Oxford Nanopore Technologies (R9.4.1 flow cells/ Kit10 chemistry).

## DISCUSSION

This study compared two sequencing technologies for the comprehensive characterization of 27 *S. pneumoniae* isolates collected from IPD patients. It additionally explored the concordance between WGS-based data and conventional typing techniques.

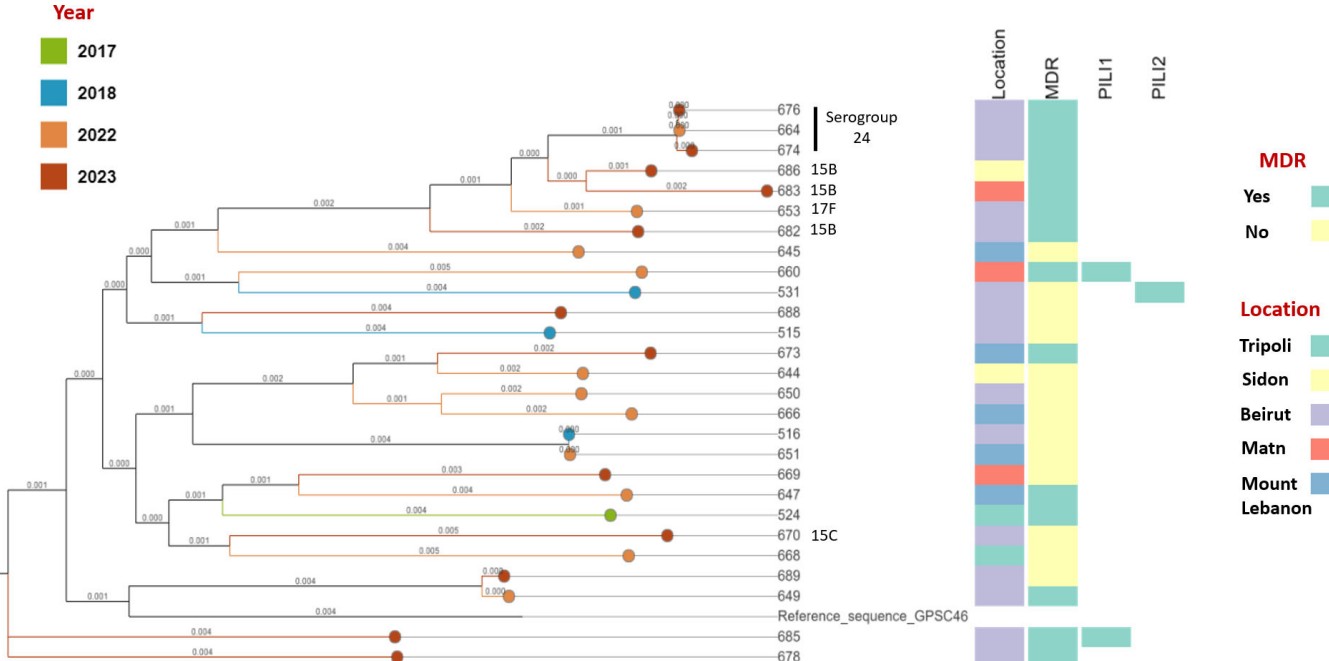

**FIG 3** Maximum-likelihood phylogenetic tree of *Streptococcus pneumoniae* isolates (*n* = 27) constructed from genome data generated by hybrid assembly of ONT_V10 and Illumina data. The tree nodes are colored by year of collection. Abbreviations: MDR, multidrug resistance; PILI1, type 1 pili; PILI2, type 2 pili.

We reported that the concordance between WGS-based serotyping and multiplex PCR was 88%, due to discrepancies at the serogroup level. Henares et al. showed that over 91% of *S. pneumoniae* isolates sequenced by ONT and Illumina were assigned with the same serotype as conventional testing (31). However, the authors used serological testing based on anticapsular antisera. In this study, we utilized multiplex PCR based on seven different reactions. S649 was identified as serotype 3 in the lab and as serotype 38 by WGS. Notably, primers for serotypes 3 and 38 are included in the same multiplex PCR. Similarly, in S653 and S670, primers for serotypes 4, 17F, 12F/12A, and 15B/15C were included in the same reaction. Subjective reading of gel results may have been implicated in the discordance between PCR and WGS serotyping results. We additionally demonstrated that the concordance between Illumina and ONT was 100% at the serogroup level.

The concordance in AMR data in this study was 100% between Illumina and ONT_V10 for penicillin, chloramphenicol, clindamycin, erythromycin, and tetracycline and 96% for trimethoprim/sulfamethoxazole. Overall, the concordance between AST results and WGS-based AMR prediction was high. It reached 100% for chloramphenicol and erythromycin and 96% for clindamycin. The lowest concordance (89%) was obtained in tetracycline, where three isolates were found resistant by WGS but sensitive by AST. ONT_V14 improved AMR data for fluoroquinolones and trimethoprim/sulfamethoxazole. Most importantly, all isolates with previously missing pbp-typing and penicillin resistance data processed by ONT_V14 had improved pbp-typing results. Interestingly, a study on Gram-negative bacteria demonstrated that ONT_V14 was accurate in detecting >94% of AMR genes with as little as 30× coverage with Flye (32).

This study shows that the assemble-Bac-ONT pipeline is practical and accurate for long-read sequencing data analysis compared to Canu. The pipeline's capacity to analyze *S. pneumoniae* genomes with limited computational resources highlights its efficiency and suitability for use in low- and middle-income countries (LMICs), where access to high-performance computing may be restricted. The pipeline utilizes Flye, which exhibits the smallest sequencing errors among long-read assemblers (33). Moreover, it polishes Flye assemblies using medaka, which was previously shown to improve single-nucleotide polymorphisms (SNPs) error rates over unpolished assemblies (34). Wick et al.

showed that Canu suffered from contiguity problems and was the slowest assembler among the assessed long-read assembly tools (33). Moreover, Mckinney et al. demonstrated that Canu assemblies displayed decreased accuracy due to indels and substitutions that affect downstream analysis (35).

Both long- and short-read FASTA files from this study were analyzed with Pathogenwatch to maximize the uniformity of results. However, the gps-unified pipeline provides a reliable, fast, and practical workflow for Illumina-only data analysis. It outputs AMR, serotyping, MLST, GPSC, virulence, and QC data.

Most isolates processed by ONT_V10 had novel STs due to missing data in MLST alleles. This phenomenon may be attributed to inaccurate assemblies within sequence typing regions, which are sensitive to variations in SNPs. Downstream analyses, such as variant calling and genome assembly accuracy, may have been affected by the reduced mean quality score initially obtained in the ONT_V10 reads. ONT_V14 improved MLST typing in 11/12 isolates. This may be attributed to the lower error rates expected from the V14 kits and R10 flow cells. Bogaerts et al. showed that the data generated from ONT R10 flow cells had very similar trends to those from Illumina, while ONT R9 data deviated slightly from it (36). The authors additionally found that the newer ONT chemistry could generate accurate and reliable SNP-based phylogenies.

Currently, hybrid assembly remains the most robust approach for bacterial genomes full reconstruction (34). Moreover, it is the optimal tool for obtaining maximum contiguity (35). We investigated two strategies for combining ONT and Illumina data in hybrid assemblies. Both resulted in superior N50 values compared to ONT-only or Illumina-only assemblies. In addition, we found that Pilon compared to Unicycler is less time-consuming and generates more contiguous genomes at ONT depth >100×. Lerminiaux et al. explored which ONT sequencing depth produced the best hybrid assembly in 12 bacterial isolates (32). The authors revealed that a minimum of 30× ONT depth is sufficient for high‑quality hybrid genome assembly with Unicycler. The ONT sequencing depth should be considered when choosing an appropriate approach for hybrid assembly.

A strong point of this study was the inclusion of isolates with varying ONT sequencing depths. This mimics clinical setting conditions, where multiplexing several samples in one flow cell is cost-effective and practical. Multiplexing is more likely to produce isolates with different sequencing depths. We additionally focused on a single bacterial organism to provide specific recommendations for optimizing its genomic characterization. Indeed, sequencing outputs do not solely rely on the technique used, but also on the bacterial species investigated (14, 34). One limitation of this study is the lack of phenotypic serotyping using anti-sera, since this technique remains the golden standard for *S. pneumoniae* serotyping. Another issue was the availability of only one personnel for the analysis of serotyping PCR results by gel electrophoresis. This may have increased the subjective interpretation risk. Moreover, four isolates lacked sufficient DNA for Illumina and were re-extracted. Ideally, the same DNA should be used in all techniques. Finally, the sample size was limited, but higher than in most similar studies (14, 32, 34, 35).

We demonstrated the reliability of the ONT V10 kit and R9.4.1 flow cells for *S. pneumoniae* identification, serotyping, GPSC, and AMR prediction. Implementing ONT sequencing in clinical settings is feasible due to its low costs, the improved accuracy by the V14 chemistry, and the high concordance between conventional diagnostics results and WGS data. The flexibility, portability, and reproducibility of the proposed pipelines further facilitate the implementation of the technique in research and clinical laboratories. Accordingly, this technique paves the way for LMICs to harness the benefits of genomics-led health care. ONT and downstream analysis are continuously improving, which may facilitate high-resolution genotyping from ONT-only data in the future (14). A recent study by Hong et al. revealed that ONT data generated by R10.4.1 flow cells exhibited accuracy that is comparable to Illumina sequences (37). The authors used the Dorado 0.5.0 Super Accurate 4.3 basecalling model, which is more accurate than the model used in this study. The utilization of super-accuracy models in this study

was limited by the availability of computing and time resources. Another study by Bogaerts et al. revealed that quality scores were improved in R10 compared to R9 data in *E. coli* and *L. monocytogenes* (36). Moreover, the authors constructed SNP-based *E. coli* phylogenies from ONT R10 or Illumina data. They demonstrated that SNP distance matrices and tree topologies were almost identical between both technologies. Another study demonstrated that genomes assembled with ONT‐only data contained less than 100 single-nucleotide variants (SNVs) compared to hybrid assemblies (32). The capacity of ONT_V14 to improve AMR prediction, MLST, and pbp typing, as demonstrated in this study, further highlights the promising potential of Nanopore technology for comprehensive bacterial sequencing and analysis.

## ACKNOWLEDGMENTS

This study was partially funded by a grant from Pfizer to G.D. (WI227709). The funders had no role in study design, data collection, and interpretation, or the decision to submit the work for publication. We thank the LIPSP collaborators for their contributions to sample collection and data acquisition.

The authors confirm contribution to the paper as follows: Supervision: G.D. and A.A.F., Study conception and design: A.A.F., F.D., and G.D.; Funding acquisition: G.D. and A.A.F.; Methodology: F.D., F.A., and J.R.J.; Clinical Coordination: N.H., A.Z., and C.B., Genomic and data analysis: F.D.; Data curation and interpretation of results: F.D., A.A.F., and G.D., Validation: A.A.F. and F.D.; Writing—draft manuscript: F.D., Writing—Review and Editing: G.D., A.A.F., F.D., J.R.J., G.A., and G.M. All authors read and approved the final version of the manuscript.

## AUTHOR AFFILIATIONS

[1]Center for Infectious Diseases Research (CIDR) and WHO Collaborating Center for Reference and Research on Bacterial Pathogens, American University of Beirut, Beirut, Lebanon
[2]Department of Experimental Pathology, Immunology, and Microbiology, Faculty of Medicine, American University of Beirut, Beirut, Lebanon
[3]Department of Pathology and Laboratory Medicine, American University of Beirut Medical Center, Beirut, Lebanon
[4]Department of Pediatrics and Adolescent Medicine, Faculty of Medicine, American University of Beirut, Beirut, Lebanon

## AUTHOR ORCIDs

Fatima Dakroub http://orcid.org/0000-0002-7432-0977
Antoine Abou Fayad http://orcid.org/0000-0002-7516-7919
Ghassan S. Dbaibo http://orcid.org/0000-0002-5813-5878

## FUNDING

| Funder | Grant(s) | Author(s) |
| --- | --- | --- |
| Pfizer (Davis) | WI227709 | Ghassan S. Dbaibo |

## AUTHOR CONTRIBUTIONS

Fatima Dakroub, Conceptualization, Data curation, Formal analysis, Investigation, Methodology, Project administration, Software, Validation, Visualization, Writing – original draft, Writing – review and editing | Fata Akl, Methodology, Writing – review and editing | Jose Rita Gerges, Methodology, Writing – review and editing | Nancy Hourani, Project administration, Writing – review and editing | Celina F. Boutros, Project administration, Writing – review and editing | George F. Araj, Resources, Writing – review and editing | Ghassan M. Matar, Resources, Supervision | Antoine Abou Fayad, Conceptualization, Data curation, Investigation, Project administration, Resources, Supervision,

Validation, Visualization, Writing – review and editing | Ghassan S. Dbaibo, Conceptualization, Funding acquisition, Investigation, Project administration, Resources, Supervision, Validation, Visualization, Writing – review and editing.

## DATA AVAILABILITY

All supporting data and protocols are provided within the article or through supplemental material.

Illumina raw reads were deposited in the European Nucleotide Archive (ENA) and can be accessed using the project number PRJEB88726. ONT reads were deposited in Sequence Read Archive (SRA) and can be accessed using the following link: https://www.ncbi.nlm.nih.gov/sra/PRJNA1253082

## ETHICS APPROVAL

This study was approved by the AUB Institutional Review Board (PED.GD.01). This is a molecular epidemiology surveillance-based study in which samples are duly anonymized; therefore, no informed consent was requested.

## ADDITIONAL FILES

The following material is available online.

### Supplemental Material

**Supplemental tables and figures (Spectrum01294-24-s0001.docx).** Tables S1 to S8, and Fig. S1 and S2.

### Open Peer Review

**PEER REVIEW HISTORY (review-history.pdf).** An accounting of the reviewer comments and feedback.

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
