## [Reviewer comments · Microbiology Spectrum]

Microbiology Spectrum

Comparison of Illumina and Oxford Nanopore Technology systems for the genomic characterization of *Streptococcus pneumoniae*

Fatima Dakroub, Fata Akl, Nancy Hourani, Jose Rita Gerges, Celina Boutros, GEORGE ARAJ, Ghassan Matar, Antoine About Fayad, and Ghassan Dbaibo

Corresponding Author(s): Ghassan Dbaibo, American University of Beirut

Review Timeline:

Submission Date:	June 7, 2024
Editorial Decision:	September 3, 2024
Revision Received:	September 28, 2024
Editorial Decision:	October 27, 2024
Revision Received:	November 23, 2024
Accepted:	January 8, 2025

Editor: Sophia Georghiou

Reviewer(s): Disclosure of reviewer identity is with reference to reviewer comments included in decision letter(s). The following individuals involved in review of your submission have agreed to reveal their identity: Xianwu Guo (Reviewer #2); Chien-Shun Chiou (Reviewer #3)

Transaction Report:

DOI: <https://doi.org/10.1128/spectrum.01294-24>

Re: Spectrum01294-24 (Comparison of Illumina and Oxford Nanopore Technology systems for the genomic characterization of *Streptococcus pneumoniae*)

Dear Prof. Ghassan Dbaibo:

Thank you for the privilege of reviewing your work. Below you will find my comments, instructions from the Spectrum editorial office, and the reviewer comments.

Revision Guidelines

Sincerely,
Sophia Georghiou
Editor
Microbiology Spectrum

Reviewer #2 (Public repository details (Required)):

27 genome sequences of the strains

Reviewer #2 (Comments for the Author):

This manuscript is to compare two methods of WGS (whole genome sequencing) in application to the serotyping and other features of clinic strains of bacterium, *S. pneumoniae*. It has some significance. However, I have some concerns on this

manuscript.

1. Please explain well the reasons why you explore the features of clinic strains by WGS methods. If the clinic strains can be isolated, the classic methods (serological methods and MIC test) can be applied much easier, fast, economical, clear and reliable. If it is not necessary to isolate and cannot be isolated, WGS such as metagenomic analysis showed some advantages.
2. When compared these two methods (ONT and Illumina) for bacterial genome sequencing, the better is to combine these two methods for genome assembly. This is a common practice. This manuscript confirmed this activity in sequencing. Thus, it showed some sense but it is novel practice.
3. Another concern is the serotyping result. Authors used the PCR and WGS methods for comparisons. I think it should be firstly identified by serological test for serotyping and then use PCR and WGS methods for comparison, which can be inferred the reliable result. The present result only showed the accordance of them but which is better, more objective?
4. As regards antibiotic resistance, authors say "Antimicrobial susceptibility testing (AST) was performed using disk diffusion to determine resistance profiles". Please give more information. For example, concentration of antibiotics was used for each test. It seems no application of MIC (minimum inhibitory concentration) for all strains because it only mentioned penicillin resistance detection with MIC. On the other hand, antibiotic resistance can be determined by unique gene but not only depended upon this gene. WGS can give information on the antibiotic resistance of strains but the real resistance profile of bacteria only can be tested by experiments hitherto.
5. Phylogenetic analysis based on the genome indicates the 2022 and 2023 strains are distinct from strains isolated in 2017 and 2018 according to the Figure 5. This result is interesting. Please give more information and good explanation. On the other hand, "isolates with simultaneous predicted resistance against penicillin, clindamycin, erythromycin, tetracycline and sulfamethoxazole clustered together"(last paragraph at the section "Results") is not clear. Please describe in detail and indicate on the Figure 5.

Reviewer #3 (Comments for the Author):

The authors conducted a comparison between Illumina and Oxford Nanopore Technologies (ONT) sequencing for identifying serotypes (serogroups), sequence types (STs), antimicrobial resistance (AMR) genetic determinants, and Global Pneumococcal Sequence Clusters (GPSCs) of *Streptococcus pneumoniae*. The study used Illumina and ONT R9 sequencing on 27 isolates, with ONT R10 applied to 12 of these isolates. While this comparison is relevant, the manuscript could benefit from a more concise presentation of the research findings.

Major Comments:

1. **Conciseness and Relevance:** The manuscript is currently too lengthy, with an excessive number of tables and figures. Consider condensing the text to focus on the most critical findings and discussions. The comparison between Illumina and ONT sequencing can be presented more straightforwardly. Streamlining the results section and moving some of the less critical data and detailed tables to the supplemental material would make the manuscript more concise and accessible to readers.
2. **Accuracy of Sequencing Technologies:** The manuscript should address the accuracy differences between ONT and Illumina sequencing more thoroughly. Although ONT sequencing is traditionally considered less accurate than Illumina, recent advancements, such as the use of the R10.4 chip and basecalling with the Dorado Sup 4.3 model, have demonstrated that ONT can achieve comparable accuracy to Illumina sequencing, as shown in studies with *Salmonella Enteritidis* and *Listeria monocytogenes* (Hong et al., 2024, *Microbiol Spectr* 12:e0050924). The manuscript should refer to these advancements and their implications for the accuracy of the ONT data used in this study.
3. **Outdated and Supplementary Data:** Some of the data presented appears outdated or may not significantly contribute to the main findings. It is recommended to critically assess the relevance of all data included. Data that do not directly support the manuscript's primary conclusions should be moved to the supplemental material or removed if they do not add substantial value to the study.
4. **Comprehensive Revision Needed:** A comprehensive revision of the manuscript's content is necessary to enhance clarity and focus. The authors should ensure that each section is aligned with the main objectives of the study and contributes directly to the key comparisons between the sequencing technologies.

Other comments:

1. Many sentences in the manuscript are difficult to understand. Please revise them for clarity and readability to ensure the content is accessible to a broader audience.
2. Many tools used in the sequence analysis, such as SPAdes, Canu, Porechop, etc. lack specific details about the versions and sources. Please include the version numbers and provide appropriate references or URLs for these tools in the Methods section.
3. The information presented in Figure 1 can be described in the Methods section. Consider removing Figure 1 from the main text or placing it in the Supplemental materials if it does not add significant value.
4. Figure 4 should also be considered for exclusion from the main text or moved to the Supplemental materials if it is not crucial to the primary findings.
5. PneumOKITy is mentioned but not described in the Methods section. Please provide a detailed description of this tool,

including its purpose and how it was used in the study.

6. In Table 1, clarify whether you have phenotypic serotype data for the 27 isolates. If phenotypic serotyping was not performed, please explain how you determined the accuracy of the serotype predictions.

7. Specify which tools within Pathogenwatch were used to predict serotypes in Table 1. This will help readers understand the basis of your serotype predictions.

8. Since PneumoKITy appears to be a useful tool, Table 1 should include serotype predictions for all isolates to provide a comprehensive comparison.

9. Table 1 should be expanded to include serotype results determined or predicted using antisera, PCR, Pathogenwatch tools, and PneumoKITy. Additionally, results from Illumina, ONT_V10, and ONT_V14 sequences should be included in the table.

10. The terminology in Table 2 needs clarification. Next-generation sequencing (NGS) is used to identify antimicrobial resistance genetic determinants (genotypes), not phenotypes. Include ONT_V14 data in Table 2 for a more comprehensive comparison.

11. Table 3 should list the predicted ST types using Illumina, ONT_V10, and ONT_V14 sequences. Consider moving the allelic data for the 7 genes to a Supplemental table.

12. The information currently in Table 4 can be integrated into Table 3 to reduce redundancy. Also, specify which basecaller was used for calling ONT_V14 reads.

13. The details provided in Figure 3 can be described within the Methods section, which may allow you to remove the figure unless it provides a unique value that cannot be conveyed in text.

14. In Figure 4, some isolates (e.g., S531, S678, S689) show poor assembly results. Discuss potential factors contributing to these results, such as sequencing depth, DNA quality, or specific isolate characteristics.

15. The statement "A maximum of 35 million base pairs were loaded into each R9.4.1 flow cell" requires clarification. Explain how the number of base pairs loaded into the flow cell was determined.

16. The sentence on lines 185-187 is unclear. Revise it for clarity. Correctly, seroBA is used for predicting serotypes, not for serotyping.

17. The description of Panaroo on lines 205-206 needs correction. Panaroo is a graph-based pangenome clustering tool (<https://pubmed.ncbi.nlm.nih.gov/32698896/>), not a tool for multiple sequence alignment. Revise the statement to accurately describe its function.

18. The statement on lines 223-224 about the ONT_V10 mean quality score is not clear. Please rephrase to clearly explain the incorrect base call probability and its implications.

The authors conducted a comparison between Illumina and Oxford Nanopore Technologies (ONT) sequencing for identifying serotypes (serogroups), sequence types (STs), antimicrobial resistance (AMR) genetic determinants, and Global Pneumococcal Sequence Clusters (GPSCs) of *Streptococcus pneumoniae*. The study used Illumina and ONT R9 sequencing on 27 isolates, with ONT R10 applied to 12 of these isolates. While this comparison is relevant, the manuscript could benefit from a more concise presentation of the research findings.

Major Comments:

1. **Conciseness and Relevance:** The manuscript is currently too lengthy, with an excessive number of tables and figures. Consider condensing the text to focus on the most critical findings and discussions. The comparison between Illumina and ONT sequencing can be presented more straightforwardly. Streamlining the results section and moving some of the less critical data and detailed tables to the supplemental material would make the manuscript more concise and accessible to readers.
2. **Accuracy of Sequencing Technologies:** The manuscript should address the accuracy differences between ONT and Illumina sequencing more thoroughly. Although ONT sequencing is traditionally considered less accurate than Illumina, recent advancements, such as the use of the R10.4 chip and basecalling with the Dorado Sup 4.3 model, have demonstrated that ONT can achieve comparable accuracy to Illumina sequencing, as shown in studies with *Salmonella Enteritidis* and *Listeria monocytogenes* (Hong et al., 2024, *Microbiol Spectr* 12:e0050924). The manuscript should refer to these advancements and their implications for the accuracy of the ONT data used in this study.
3. **Outdated and Supplementary Data:** Some of the data presented appears outdated or may not significantly contribute to the main findings. It is recommended to critically assess the relevance of all data included. Data that do not directly support the manuscript's primary conclusions should be moved to the supplemental material or removed if they do not add substantial value to the study.
4. **Comprehensive Revision Needed:** A comprehensive revision of the manuscript's content is necessary to enhance clarity and focus. The authors should ensure that each section is aligned with the main

objectives of the study and contributes directly to the key comparisons between the sequencing technologies.

Other comments:

1. Many sentences in the manuscript are difficult to understand. Please revise them for clarity and readability to ensure the content is accessible to a broader audience.
2. Many tools used in the sequence analysis, such as SPAdes, Canu, Porechop, etc. lack specific details about the versions and sources. Please include the version numbers and provide appropriate references or URLs for these tools in the Methods section.
3. The information presented in Figure 1 can be described in the Methods section. Consider removing Figure 1 from the main text or placing it in the Supplemental materials if it does not add significant value.
4. Figure 4 should also be considered for exclusion from the main text or moved to the Supplemental materials if it is not crucial to the primary findings.
5. PneumoKITy is mentioned but not described in the Methods section. Please provide a detailed description of this tool, including its purpose and how it was used in the study.
6. In Table 1, clarify whether you have phenotypic serotype data for the 27 isolates. If phenotypic serotyping was not performed, please explain how you determined the accuracy of the serotype predictions.
7. Specify which tools within Pathogenwatch were used to predict serotypes in Table 1. This will help readers understand the basis of your serotype predictions.
8. Since PneumoKITy appears to be a useful tool, Table 1 should include serotype predictions for all isolates to provide a comprehensive comparison.
9. Table 1 should be expanded to include serotype results determined or predicted using antisera, PCR, Pathogenwatch tools, and PneumoKITy. Additionally, results from Illumina, ONT_V10, and ONT_V14 sequences should be included in the table.
10. The terminology in Table 2 needs clarification. Next-generation sequencing (NGS) is used to identify antimicrobial resistance genetic determinants (genotypes), not phenotypes. Include ONT_V14 data in Table 2 for a more comprehensive comparison.
11. Table 3 should list the predicted ST types using Illumina, ONT_V10, and ONT_V14 sequences. Consider moving the allelic data for the 7 genes to a

Supplemental table.

12. The information currently in Table 4 can be integrated into Table 3 to reduce redundancy. Also, specify which basecaller was used for calling ONT_V14 reads.
13. The details provided in Figure 3 can be described within the Methods section, which may allow you to remove the figure unless it provides a unique value that cannot be conveyed in text.
14. In Figure 4, some isolates (e.g., S531, S678, S689) show poor assembly results. Discuss potential factors contributing to these results, such as sequencing depth, DNA quality, or specific isolate characteristics.
15. The statement "A maximum of 35 million base pairs were loaded into each R9.4.1 flow cell" requires clarification. Explain how the number of base pairs loaded into the flow cell was determined.
16. The sentence on lines 185-187 is unclear. Revise it for clarity. Correctly, seroBA is used for predicting serotypes, not for serotyping.
17. The description of Panaroo on lines 205-206 needs correction. Panaroo is a graph-based pangenome clustering tool (<https://pubmed.ncbi.nlm.nih.gov/32698896/>), not a tool for multiple sequence alignment. Revise the statement to accurately describe its function.
18. The statement on lines 223-224 about the ONT_V10 mean quality score is not clear. Please rephrase to clearly explain the incorrect base call probability and its implications.

Dear Dr. Sophia Georghiou,

We would like to submit our revised manuscript “Comparison of Illumina and Oxford Nanopore Technology systems for the genomic characterization of *Streptococcus pneumoniae*”.

We appreciate the insightful comments made by the editor and reviewers. We thank them for the time and effort spent reviewing the manuscript.

We have carefully read the reviewers’ comments and responded to them individually, indicating how we addressed each comment and describing the changes amended.

The revisions have been approved by all authors. The changes are marked in tracking mode in the manuscript. Our responses to the referee’s comments are included below.

We hope that the revisions will meet the reviewers’ approval.

Reviewer #2 (Comments for the Author):

This manuscript is to compare two methods of WGS (whole genome sequencing) in application to the serotyping and other features of clinic strains of bacterium, *S. pneumoniae*. It has some significance. However, I have some concerns on this manuscript.

1. Please explain well the reasons why you explore the features of clinic strains by WGS methods. If the clinic strains can be isolated, the classic methods (serological methods and MIC test) can be applied much easier, fast, economical, clear and reliable. If it is not necessary to isolate and cannot be isolated, WGS such as metagenomic analysis showed some advantages.

Response: The practice of clinical bacteriology is projected to be revolutionized by WGS. Indeed, sequencing can characterize bacterial pathogens directly from clinical specimens. In many developing countries, antibiotics are available over-the-counter without prescription. This leads to antibiotic overuse in the community and frequently interferes with the ability to culture *S. pneumoniae* from patients with pneumococcal infections who have already received antibiotics. WGS in this regard offers the ability to identify these bacteria even without a positive culture providing the necessary information for clinical management (e.g. antimicrobial resistance-AMR) and for epidemiological impact (e.g. serotype identification relative to pneumococcal vaccine use). Although currently not as economical as standard culture methods, the decreasing cost over time will render WGS more competitive in the near future. Indeed, it is interesting to assess the WGS technologies applied directly to clinical samples. In this study, however, we received *S. pneumoniae* isolated from blood or CSF that were cultured on agar plates for clinical assessment.

DNA extraction is the first common step between multiplex PCR intended for serotyping and WGS. Library preparation by ONT typically takes between 1.5 and 2.5 hours, depending on the experience of the laboratory technician. Thus, sequencing can be performed on the same day of receiving the sample. With the real-time signaling feature of the MinKnow software, data is generated simultaneously during sequencing. Hence, when enough reads are obtained to produce a sufficient sequencing depth (the cutoff is 30x and above at our lab), AMR and serotyping

results can be generated confidently. This is demonstrated in this work, in which isolates at a sequencing depth of 37X, 39.9X, 41X ext. had high concordance with Illumina and AST data. The V14 chemistry from ONT, which replaced the V10 chemistry, further improves sequencing accuracy and decreases the time needed to generate raw reads. Usually, 24 hours of sequencing on the Mk1c or Mk1b devices are enough to produce sufficient reads for analysis.

On the other hand, AST requires subculturing the bacteria on MH blood agar plates, after which the antibiotic disks are added the next day. Then, the results are interpreted the day after. Hence AST requires at least 48 hours to generate AMR results. Multiplex PCR is also time-consuming, since it is not cost-effective to run the seven reactions simultaneously. Each PCR run takes three hours to be completed (excluding preparation time). Then the PCR products are visualized by gel electrophoresis after each reaction. Moreover, some isolates are not typeable due to the lack of specific primers in the PCR reactions.

In addition to AMR and serotypes, WGS generates a wealth of information, including MLST, GPSCs, pbp typing and virulence genetic markers. Moreover, outbreaks in clinical settings are accurately assessed by WGS-based phylogenies. ONT sequencing is much cheaper than Illumina, and it is expected that sequencing costs will continue to fall.

We do not claim that WGS should fully replace conventional microbiology diagnostics. However, implementing it in the clinical setting will add value to the capacity of physicians in bed-side decision making. For example, if a bacterium is resistant to antibiotic X by WGS, but susceptible by culture, should a physician be comfortable administering the drug to the patient? What if the gene was present in the genome but not expressed during AST testing?

An interesting study by Sauerborn et al. demonstrated the value of sequencing in a complex infection case. The authors showed that ONT can accurately detect low-abundance plasmid-mediated resistance in *Klebsiella pneumoniae*, which often remains undetected by conventional methods (1). Moreover, *Streptococcus mitis* and *Streptococcus oralis* are commonly misidentified as *S. pneumoniae* by conventional culture. WGS can aid in resolving this issue.

2. When compared these two methods (ONT and Illumina) for bacterial genome sequencing, the better is to combine these two methods for genome assembly. This is a common practice. This manuscript confirmed this activity in sequencing. Thus, it showed some sense but it is novel practice.

Thank you for this comment. Indeed, this is the first manuscript to offer a comprehensive and detailed comparison between different sequencing technologies and conventional diagnostics in a considerable number of isolates belonging to the same species. This allowed us to provide specific recommendations for optimizing the genomic characterization of *S. pneumoniae*. The evaluation of hybrid assembly approaches added significant value to our findings, further enhancing the depth and practical relevance of our study.

The study offers a guide for clinical and research laboratories seeking to adopt bacterial sequencing, by providing important considerations when choosing sequencing platforms and analysis pipelines. It is especially relevant for low and middle-income countries (LMICs) that

may have restricted access to computation resources and Illumina sequencing, which is relatively expensive.

3. Another concern is the serotyping result. Authors used the PCR and WGS methods for comparisons. I think it should be firstly identified by serological test for serotyping and then use PCR and WGS methods for comparison, which can be inferred the reliable result. The present result only showed the accordance of them but which is better, more objective?

Response:

Phenotypic serotyping data was not available for isolates in this study.

Multiplex PCR and serological tests demonstrate comparable accuracy, but both can suffer from subjective interpretation of their results. Please refer to the answer of comment 6 (posted by reviewer 3) for more details.

The lack of phenotypic serotyping data was acknowledged in the Discussion section as follows:

Page 18, Lines 422-424: {One limitation of this study is the lack of phenotypic serotyping using anti-sera, since this technique remains the golden standard for *S. pneumoniae* serotyping.}

4. As regards antibiotic resistance, authors say "Antimicrobial susceptibility testing (AST) was performed using disk diffusion to determine resistance profiles". Please give more information. For example, concentration of antibiotics was used for each test. It seems no application of MIC (minimum inhibitory concentration) for all strains because it only mentioned penicillin resistance detection with MIC. On the other hand, antibiotic resistance can be determined by unique gene but not only depended upon this gene. WGS can give information on the antibiotic resistance of strains but the real resistance profile of bacteria only can be tested by experiments hitherto.

Response:

A. The AST disk diffusion experiments and the interpretation of results were performed according to the CLSI guidelines, which also provide instructions on the disk contents. The appropriate reference for the guidelines was added in the Materials and Methods section as follows:

Page 7, Lines 176-178: {Antimicrobial susceptibility testing (AST) was performed using disk diffusion to determine resistance profiles against chloramphenicol, clindamycin, erythromycin, sulfamethoxazole/trimethoprim, tetracycline and levofloxacin following the CLSI guidelines (19).}

B. Thank you for this comment. We agree that the resistance profiles of bacteria can exhibit discrepancies at the genotypic and phenotypic levels. The penicillin resistance data as determined by AST in the lab was added to Table S5 in the supplemental materials. The AST data for two isolates was missing. The penicillin results of the remaining 25 isolates were 100% concordant between AST and WGS.

However, we avoided comparing quantitative MIC values between AST and WGS. The latter often predicts resistance qualitatively (resistant or susceptible) but struggles to predict precise MIC values. The exact MIC levels relationship with specific genetic mutations is not always straightforward. Moreover, MIC quantitative predictions depend heavily on the quality and comprehensiveness of the reference database utilized.

5. Phylogenetic analysis based on the genome indicates the 2022 and 2023 strains are distinct from strains isolated in 2017 and 2018 according to the Figure 5. This result is interesting. Please give more information and good explanation. On the other hand, "isolates with simultaneous predicted resistance against penicillin, clindamycin, erythromycin, tetracycline and sulfamethoxazole clustered together"(last paragraph at the section "Results") is not clear. Please describe in detail and indicate on the Figure 5.

Response:

A. All isolates collected in 2017/2018 share a most recent common ancestor with a corresponding isolate collected in 2022/2023. For example:

S531 (2018) and 660 (2022) share the same branch point.

S515 (2018) and S688 (2023) also share the same branch point.

S516 (2018) and S651 (2022) are located on the same branch with 0.0 horizontal distance between them.

Moreover, the direct most recent common ancestor is not the same between all isolates collected in 2017/2018.

Hence, we do not observe a clear distinction between isolates from 2017/2018 compared to those from 2022/2023.

B. In this study, isolates were assigned as MDR if they were predicted to be resistant against at least one antimicrobial drug in three or more antimicrobial categories. These predictions were obtained from hybrid assemblies.

Some of the MDR isolates exhibited more antimicrobial resistance markers than other isolates within the same MDR group. We did not want to further divide the MDR group into subgroups based on the number of predicted resistance markers in each isolate. We did not consider this to be highly relevant to the objectives of this study.

That said, we found it interesting that isolates of serogroup 24 or serotype 15B exhibited identical AMR predictions. Our findings, showing that serotype 15B and serogroup 24/ST15253 causing IPD recently in Lebanon are MDR, provide valuable insights for public health officials and researchers focused on *S. pneumoniae* in the country. Thus, we briefly highlighted that these isolates had resistance genotypes that infer simultaneous resistance to penicillin, clindamycin, erythromycin, tetracycline, and sulfamethoxazole. The serogroups/serotypes of these isolates are already clearly indicated in the figure.

Reviewer #3 (Comments for the Author):

The authors conducted a comparison between Illumina and Oxford Nanopore Technologies (ONT) sequencing for identifying serotypes (serogroups), sequence types (STs), antimicrobial resistance (AMR) genetic determinants, and Global Pneumococcal Sequence Clusters (GPSCs) of *Streptococcus pneumoniae*. The study used Illumina and ONT R9 sequencing on 27 isolates, with ONT R10 applied to 12 of these isolates. While this comparison is relevant, the manuscript could benefit from a more concise presentation of the research findings.

Major Comments:

1. Conciseness and Relevance: The manuscript is currently too lengthy, with an excessive number of tables and figures. Consider condensing the text to focus on the most critical findings and discussions. The comparison between Illumina and ONT sequencing can be presented more straightforwardly. Streamlining the results section and moving some of the less critical data and detailed tables to the supplemental material would make the manuscript more concise and accessible to readers.

Response: We thank the reviewer for their insight and helpful critiques regarding our manuscript. We have made several edits to address all concerns and suggestions raised. The manuscript was reviewed thoroughly for conciseness and clarity and the number of tables and figures was reduced.

*2. Accuracy of Sequencing Technologies: The manuscript should address the accuracy differences between ONT and Illumina sequencing more thoroughly. Although ONT sequencing is traditionally considered less accurate than Illumina, recent advancements, such as the use of the R10.4 chip and basecalling with the Dorado Sup 4.3 model, have demonstrated that ONT can achieve comparable accuracy to Illumina sequencing, as shown in studies with *Salmonella Enteritidis* and *Listeria monocytogenes* (Hong et al., 2024, *Microbiol Spectr* 12:e0050924). The manuscript should refer to these advancements and their implications for the accuracy of the ONT data used in this study.*

Response: A review of the improvements in the ONT technology using R10.4.1 flow cells was added to the Discussion section as follows:

Page 19, Lines 433-447: {ONT and downstream analysis are continuously improving, which may facilitate high-resolution genotyping from ONT-only data in the future (13). A recent study by Hong et al. revealed that ONT data generated by R10.4.1 flow cells exhibited accuracy that is comparable to Illumina sequences (33). The authors used the Dorado 0.5.0 Super Accurate 4.3 basecalling model, which is more accurate than the model used in this study. The utilization of super-accuracy models in this study was limited by the availability of computing and time resources. Another study by Bogaerts et al. revealed that quality scores were improved in R10 compared to R9 data in *E. coli* and *L. monocytogenes* (32). Moreover, the authors constructed SNP-based *E. coli* phylogenies from ONT R10 or Illumina data. They demonstrated that SNP distance matrices and tree topologies were almost identical between both technologies. Another

study demonstrated that genomes assembled with ONT-only data contained less than 100 single nucleotide variants (SNVs) compared to hybrid assemblies (28). The capacity of ONT_V14 to improve AMR prediction, mlst and pbp typing, as demonstrated in this study, further highlights the promising potential of Nanopore technology for comprehensive bacterial sequencing and analysis.}

3. Outdated and Supplementary Data: *Some of the data presented appears outdated or may not significantly contribute to the main findings. It is recommended to critically assess the relevance of all data included. Data that do not directly support the manuscript's primary conclusions should be moved to the supplemental material or removed if they do not add substantial value to the study.*

Response: We re-examined the results to include only those that are relevant to the main findings of the study. The Results section in the main text was modified accordingly. Moreover, Figures 1 and 4 were moved to supplemental materials and Tables 3 and 4 were merged to reduce redundancy.

4. Comprehensive Revision Needed: *A comprehensive revision of the manuscript's content is necessary to enhance clarity and focus. The authors should ensure that each section is aligned with the main objectives of the study and contributes directly to the key comparisons between the sequencing technologies.*

Response: We thank the reviewer for pointing out this issue and appreciate the opportunity to further refine the manuscript. We have revised the entire manuscript to improve clarity and coherence.

Other comments:

1. *Many sentences in the manuscript are difficult to understand. Please revise them for clarity and readability to ensure the content is accessible to a broader audience.*

2. *Many tools used in the sequence analysis, such as SPAdes, Canu, Porechop, etc. lack specific details about the versions and sources. Please include the version numbers and provide appropriate references or URLs for these tools in the Methods section.*

Response:

A. We understand that some sections are difficult to understand for a broader audience. We have edited many sentences in the main text and hope that they now clearly reflect the study's findings for readers.

B. The versions and sources of the pipelines and tools used for analysis are now included in the Materials and Methods section, lines 201-240.

We did not add the sources of tools that were included in pipelines (ex. Flye, SPAdes...) because these were not installed individually. We added the URLs of the pipelines instead.

3. The information presented in Figure 1 can be described in the Methods section. Consider removing Figure 1 from the main text or placing it in the Supplemental materials if it does not add significant value.

4. Figure 4 should also be considered for exclusion from the main text or moved to the Supplemental materials if it is not crucial to the primary findings.

Response: We thank you for this comment which helped make the manuscript more concise. We moved both Figures 1 and 4 to supplemental materials. They are now called Figures S1 and S2 respectively. Sub-sections 2.1 and 2.4 in the Materials and Methods section were modified to include information presented in Figure S1.

5. PneumoKITy is mentioned but not described in the Methods section. Please provide a detailed description of this tool, including its purpose and how it was used in the study.

Response: We have added a description of PneumoKITy in the Materials and Methods section as follows:

Page 10, Lines 222-225: {The serotypes of all isolates were additionally predicted using PneumoKITy (20), which is a sensitive k-mer based tool that allows the prediction of mixed *S. pneumoniae* serotypes. PneumoKITy was used to predict the serotypes of all isolates sequenced with Illumina, ONT_V10 and ONT_V14.}

6. In Table 1, clarify whether you have phenotypic serotype data for the 27 isolates. If phenotypic serotyping was not performed, please explain how you determined the accuracy of the serotype predictions.

Response: Phenotypic serotyping data is not available for isolates in this study. Serological testing of *Streptococcus pneumoniae* based on anticapsular antisera is not a standard method at our laboratory. It is an expensive and labor-intensive experiment, so we chose to substitute it with molecular typing, which exhibits high sensitivity.

The sensitivity of molecular typing by PCR was investigated by Brito et al. who showed that the results of *S. pneumoniae* isolates (n=294) typed by PCR agreed fully with those obtained by conventional immunological methods (2). Similarly, Zhou et al. demonstrated that serotyping by multiplex PCR is highly sensitive, with only 1.7% of isolates misidentified (3). Another study showed that the multiplex PCR result agreed with that of the Quellung reaction in 1,733/1750 (99%) *S. pneumoniae* isolates (4).

A major pitfall of the PCR method is the inability to type isolates that express serotypes for which no specific primers are included in the multiplex scheme. While serotyping by PCR primarily is effective and precise, Quellung testing remains the golden standard for *S. pneumoniae* serotyping, because it accounts for discrepancies between the genotypic and phenotypic serotype of an isolate.

The lack of phenotypic serotyping data was acknowledged in the Discussion section as follows:

Page 18, Lines 422-424: {One limitation of this study is the lack of phenotypic serotyping using anti-sera, since this technique remains the golden standard for *S. pneumoniae* serotyping.}

7. Specify which tools within Pathogenwatch were used to predict serotypes in Table 1. This will help readers understand the basis of your serotype predictions.

Response: The following information on Pathogenwatch was added in the Material and Methods section:

Page 9, Line 213-221: {FASTA files obtained from both pipelines were analyzed using Pathogenwatch (<https://pathogen.watch/>) to obtain final serotyping, AMR, MLST and Global Pneumococcal Sequence Clusters (GPSC) predictions. Pathogenwatch utilizes SeroBA, a k-mer-based method, to predict the serotypes of *S. pneumoniae*. It employs the PAARSNP software to infer the presence or absence of resistance genetic markers by using BLAST against a custom AMR database. Moreover, it performs Penicillin-binding proteins (pbp) typing based on the method published by Li et al. (20). The predicted resistance profiles are then interpreted using the CLSI guidelines. Pathogenwatch predicts sequence types (STs) using the MLST scheme hosted in the PubMLST database.}

Moreover, the following sentence was added as a footnote in Table 1:

{*Pathogenwatch is a platform with a user-friendly interface that utilizes SeroBA, a k-mer-based method, to predict the serotypes of *S. pneumoniae* bacteria.}

8. Since PneumoKITy appears to be a useful tool, Table 1 should include serotype predictions for all isolates to provide a comprehensive comparison.

Response: We thank the reviewer for this comment, which indeed facilitates a more comprehensive comparison of the serotypes data. PneumoKITy predictions were added for all isolates in Table 1.

9. Table 1 should be expanded to include serotype results determined or predicted using antisera, PCR, Pathogenwatch tools, and PneumoKITy. Additionally, results from Illumina, ONT_V10, and ONT_V14 sequences should be included in the table.

Response: Predicted serotypes by pathogen watch and PneumoKITy across all techniques including Illumina, ONT_V10, and ONT_V14 are now included in Table 1. Multiplex PCR data is already included for all isolates in Table 1, column 2.

Unfortunately, phenotypic serotyping data is not available for isolates in this study. Hence, serotyping results determined using antisera can't be included in Table 1.

10. The terminology in Table 2 needs clarification. Next-generation sequencing (NGS) is used to identify antimicrobial resistance genetic determinants (genotypes), not phenotypes. Include ONT_V14 data in Table 2 for a more comprehensive comparison.

Response:

A. The terminology in Table 2 was modified as follows:

- Title of the table: {**Table 2:** Predicted resistance profiles by the different sequencing technologies compared to AST results in *S. pneumoniae* isolates (n=27) causing IPD.}

- The following sentence was added to the footnote of the table:

{Pathogenwatch was used to predict antimicrobial resistance profiles in WGS-based data. Pathogenwatch utilizes the PAARSNP software to infer the presence or absence of resistance genetic markers by using BLAST against a custom AMR database.}

B. Data from ONT_V14 is now included in the table.

11. Table 3 should list the predicted ST types using Illumina, ONT_V10, and ONT_V14 sequences. Consider moving the allelic data for the 7 genes to a Supplemental table.

12. The information currently in Table 4 can be integrated into Table 3 to reduce redundancy. Also, specify which basecaller was used for calling ONT_V14 reads.

Response:

A. We kept the allelic data in Table 3 to present the MLST loci with the highest number of novel predictions in ONT_V10 [spi, 20/27 (74%) followed by xpt 12/27 (44%)]. Moreover, we aim to provide insights on how ONT_V14 improves MLST predictions at each locus. If the reviewer considers removing the alleles necessary, we can modify the table again to include only STs and the number of assigned alleles.

B. The MLST data from ONT_V14 was added to Table 3 to reduce redundancy as the reviewer suggested.

C. We added information on the base callers used for both ONT_V10 and ONT_V14 in the Materials and Methods section as follows:

Page 8, Lines 186-187: {Guppy, a base caller integrated within the MinKNOW software, was utilized for the live base calling of ONT_V10 reads using the FAST base calling model.}

Page 8, Lines 198-199: {Live base calling of ONT_V14 reads was performed using Dorado (v7.1.4) and the FAST base calling model.}

13. The details provided in Figure 3 can be described within the Methods section, which may allow you to remove the figure unless it provides a unique value that cannot be conveyed in text.

Response: We appreciate the reviewer's suggestion, but we prefer to keep this figure in the main manuscript as it visually summarizes both hybrid assembly approaches in a clear and concise manner. This makes it easier for readers to grasp the key differences between the approaches. It shows the types of input files (FASTA vs FastQ) used in each approach and details the steps involved in short-read polishing. Moreover, the figure presents an easy-to-follow summary of the findings presented in the final panel. The mentioned figure is now named "Figure 2" in the revised manuscript.

14. In Figure 4, some isolates (e.g., S531, S678, S689) show poor assembly results. Discuss potential factors contributing to these results, such as sequencing depth, DNA quality, or specific isolate characteristics.

Response: The assembly graphs in the mentioned figure were generated by hybrid assembly using Unicycler. Unlike Unicycler, Pilon does not perform assembly. Instead, it polishes the ONT assembly (FASTA) using Illumina reads. Hence it does not generate GFA files for visualization in Bandage.

- S689 analyzed with Pilon demonstrated a favorable profile (Table S1 in supplemental materials) generating a genome with only five contigs and a high N50 (1,520,944). On the other hand, S689 was assembled into 45 contigs by Unicycler and showed poor assembly results in Figure 4 (now called Figure S2). We recommended Pilon instead of Unicycler for the hybrid assembly of isolates with a sequencing depth > 100X. The initial ONT sequencing depth of S689 was high (253X), which contributed to favorable assembly QC parameters with Pilon (contig #, genome size, N50 and GC content).
- To explain the results of S687, we will compare it to S686, which had the exact ONT sequencing depth (42X). Both produced more favorable profiles in Unicycler compared to Pilon (Table S8). Moreover, both isolates had the same coverage in Illumina (~65X). Unicycler (hybrid assembly) assembled S678 and S686 into 46 and 36 contigs respectively. However, the assembly graph of S686 was much better than that of S678. We hypothesize that in S678, the coverage was low at positions that would have allowed adjacent contigs to be joined together.
The most pronounced difference between S678 and S686 was the mean Illumina quality score, which was 35 in S686 and 32.8 in S678. Although both quality scores are high, the difference could have affected downstream assembly. Interestingly, S686 and S687 were assembled into 34 and 67 contigs respectively using Illumina-only data (Table S4 in supplemental materials).
- S531 had the lowest sequencing depth in ONT among all isolates and was assembled into 276 contigs with ONT-only data (Tables S3 and S4 in supplementary materials). The low coverage of long reads, which were intended to scaffold the short-read assembly graph in Unicycler, likely hindered the resolution of complex genomic regions such as repetitive sequences. This may have led to the generation of disjointed contigs, as the long reads were unable to span the complex genomic regions and facilitate the connection of adjacent contigs, ultimately impacting the overall genome assembly quality. Consequently, the assembly remained fragmented, with unresolved sections that require additional coverage of longer reads for accurate reconstruction.

15. The statement "A maximum of 35 million base pairs were loaded into each R9.4.1 flow cell" requires clarification. Explain how the number of base pairs loaded into the flow cell was determined.

Response: The total combined genome size of the bacterial samples loaded into an R9.4.1 flow cell should not exceed 35 million base pairs (bp). This limit was established through previous in-house experiments, which concluded that exceeding this combined genome size can negatively impact the total reads output.

Unlike Illumina, we could not find a calculator that accounts for genome size, expected coverage and flow cell data output (Gb) to determine the number of isolates/run.

Here is an example of the bacterial isolates included in one of the ONT sequencing runs:

Sample	Organism	Genome size (bp)
S655	Streptococcus pneumoniae	2,200,000
S660	Streptococcus pneumoniae	2,200,000
S670	Streptococcus pneumoniae	2,200,000
S666	Streptococcus pneumoniae	2,200,000
S667	Streptococcus pneumoniae	2,200,000
S669	Streptococcus pneumoniae	2,200,000
S678	Streptococcus pneumoniae	2,200,000
S682	Streptococcus pneumoniae	2,200,000
S683	Streptococcus pneumoniae	2,200,000
S686	Streptococcus pneumoniae	2,200,000
S688	Streptococcus pneumoniae	2,200,000
PSA	Pseudomonas aeruginosa	7,500,000
Total combined genome size		31,700,000

In the above case, the total combined genome size (31,700,000 bp) was lower than the chosen cutoff (35,000,000 bp).

Note: S655 was identified in downstream analysis as *Streptococcus oralis*, with a genome size of 2,322,699 bp. S667 was identified as *Brevibacillus invocatus* with a genome size of 6,502,171 bp.

The following modification in wording was introduced to the Materials and Methods section as follows:

Page 8, Lines 183-184: {The total combined genome size of the bacterial samples loaded into each R9.4.1 flow cell (FLO-MIN106) was limited to 35 million base pairs.}

16. The sentence on lines 185-187 is unclear. Revise it for clarity. Correctly, seroBA is used for predicting serotypes, not for serotyping.

Thank you for this comment. The sentence was amended in the revised manuscript as follows:

Page 9, Lines 204-206: {The gps-unified pipeline predicts pneumococcal serotypes and lineages using SeroBA (v1.0.5) and PopPUNK (v2.6.0) respectively.}

17. The description of Panaroo on lines 205-206 needs correction. Panaroo is a graph-based pangenome clustering tool (<https://pubmed.ncbi.nlm.nih.gov/32698896/>), not a tool for multiple sequence alignment. Revise the statement to accurately describe its function.

Response: Indeed, the software Panaroo was developed for the analysis of bacterial pangenomes. Panaroo can be utilized to generate a multiple sequence alignment of the core

genome by identifying orthologous sequences in a set of sequences. We chose Panaroo because it accounts for errors introduced during bacterial genome annotation.

Below is the **command used** in our analysis:

```
panaroo \  
  --input annotations/*.gff reference/ Reference_sequence_GPSC46.gff3 \  
  --out_dir Panaroo \  
  --clean-mode strict \  
  --alignment core \  
  --core_threshold 0.98 \  
  --remove-invalid-genes \  
  --threads 8
```

The output of interest was: core_gene_alignment_filtered.aln which represents the alignment.

The --alignment option allowed us to produce an alignment of core genes instead of a pangenome alignment. Since we wanted to account for factors such as horizontal gene transfer, gene loss and gene duplication we preferred to generate an alignment that focuses on the core genome.

Moreover, we used the ‘strict’ mode because including rare plasmids was less important for the downstream task, which is building a phylogeny.

The text in the manuscript was modified as follows:

Page 10, Lines 236-238: {A core genome alignment of the *S. pneumoniae* sequences and a reference genome (GPSC46) from the global pneumococcal sequencing project was generated using Panaroo (v1.3.4) (24).}

18. The statement on lines 223-224 about the ONT_V10 mean quality score is not clear. Please rephrase to clearly explain the incorrect base call probability and its implications.

Response: A more thorough description of the quality scores and their implications was added as follows:

Page 11, Lines 248-257: {The quality scores assessed in Figure 1D correspond to Phred scores, which quantify the error probability in a base call. These scores reflect the level of confidence in the assignment of a base call by the sequencer. For example, a phred score of 30 is equivalent to the probability of an incorrect base call in 1 out of 1,000 bases in the sequencing reads.

This study reveals that the average quality score calculated from Illumina sequences (n= 27; Mean, 34.37; SD, 0.88) was higher than that of ONT_V10 (n= 27; Mean, 18.37; SD, 0.68). This indicates that the accuracy of the base sequence generated by ONT_V10 is reduced compared to

Illumina. The lower mean quality score in the ONT_V10 reads may affect downstream analyses, such as variant calling and genome assembly accuracy. To improve the accuracy of long-read assemblies, the sequencing errors can be corrected using nanopore-based polishing tools such as medaka.}

References:

1. Sauerborn E, Corredor NC, Reska T, Perlas A, Vargas da Fonseca Atum S, Goldman N, et al. Detection of hidden antibiotic resistance through real-time genomics. *Nat Commun.* 2024 Jun 28;15(1):5494.
2. Brito DA, Ramirez M, de Lencastre H. Serotyping *Streptococcus pneumoniae* by multiplex PCR. *J Clin Microbiol.* 2003 Jun;41(6):2378–84.
3. Zhou ML, Wang ZR, Li YB, Kudinha T, Wang J, Wang Y, et al. Rapid identification of *Streptococcus pneumoniae* serotypes by *cpsB* gene-based sequencing combined with multiplex PCR. *J Microbiol Immunol Infect Wei Mian Yu Gan Ran Za Zhi.* 2022 Oct;55(5):870–9.
4. Richter SS, Heilmann KP, Dohrn CL, Riahi F, Diekema DJ, Doern GV. Evaluation of pneumococcal serotyping by multiplex PCR and quellung reactions. *J Clin Microbiol.* 2013 Dec;51(12):4193–5.

Re: Spectrum01294-24R1 (Comparison of Illumina and Oxford Nanopore Technology systems for the genomic characterization of *Streptococcus pneumoniae*)

Dear Prof. Ghassan Dbaibo:

Thank you for the privilege of reviewing your work. Below you will find my comments, instructions from the Spectrum editorial office, and the reviewer comments.

Revision Guidelines

Sincerely,
Sophia Georghiou
Editor
Microbiology Spectrum

Reviewer #4 (Public repository details (Required)):

Genome assemblies or raw fastq data should be deposited in a public repository such as ENA, SRA.

Reviewer #4 (Comments for the Author):

In this manuscript, the authors perform an extensive and useful comparison of different second and third-generation sequencing technologies as well as different bioinformatic pipelines for pneumococcal genome assembly and its genomic characterization.

The manuscript is of high interest to pneumococcal surveillance programs, in the light of the actual context of switching from classic phenotypic and molecular methods to whole genome sequencing methods. The sequencing chemistry and bioinformatic programs are continuously improving to better reflect the real biological sequences of the pathogens studied, and literature has to reflect these changes. This manuscript is another evidence that supports the use of ONT sequencing for its implementation in the rapid *S. pneumoniae* genomic characterization in a cost-effective and flexible way, without the need of important computer resources. However, the latter should be better reflected on the manuscript since it is an important conclusion. In addition, specific attention should be paid throughout the manuscript to avoid mixing results with interpretation and conclusions. Finally, there is a great amount of data and the manuscript should benefit from a more concise presentation. Specific questions to address are found below:

1. Line 77. Suggest to replace ONT kit v14 by ONT_V14
2. Line 87 and 88. Suggest to replace the most in-depth by an in-depth.
3. Line 76. Typo in *S. pneumonia*
4. Line 95. Phenotypic testing is not performed in this paper, so this affirmation does not seem correct on the importance section.
5. Line 123-124, line 136. Please, replace diagnostic by typing. In addition, all disadvantages do not apply to all conventional typing methods. Please, this sentence should be reformulated.
6. Line 126. Regarding this affirmation: Illumina entails high costs of sample processing... Illumina is one of the most competitive sequencing platforms in terms of accuracy and price. The problem relies on the need to accumulate great number of samples to achieve such low cost, which increases in time-to-results. This sentence should be clarified.
7. Line 176. ¿Why beta-lactam antibiotics were not tested? In Table S5 there are indicated AST results from penicillin.
8. Line 202-242 Why to compare specifically porechop/canu and assembleBAC-ONT and not others. ¿Which is the interest? ¿Are there differences that make them interesting to compare? The manuscript should benefit from a brief explanation.
9. Line 201. There is a high level of confusion on this section. There are results such as MLST, resistance genes, serotype, etc. obtained by both the gps-unified pipeline and Pathogenwatch, but only the latter from Pathogenwatch are considered. This section should be simplified. The manuscript should benefit from just mention the pipelines and tools used within these pipelines until the obtention of the assembled genome, which is the input to Pathogenwatch.
10. Line 251-253. This is very theoretic and could be added to supplementary material.
11. Line 255-259. This is a mix of interpretation and discussion, therefore it should be better placed in other section
12. Line 263. The manuscript mentions: "canu may have generated over-assembled genomes containing an overrepresentation of repeated region". This sentence is more appropriate in the discussion section. Moreover, the interpretation may be not supported with the data provided. There are several reasons that can impact genome contiguity, such as the choice of parameters of minimum overlap length or k-mer size, the way they are handling repetitive regions, different abilities to correct sequencing errors (sequencing errors can be interpreted as new real biological sequences), how to treat genome regions with low coverage (some assemblers may break contigs in areas with low coverage to avoid uncertainty, leading to more contigs), and several additional reasons. Additional analyses will be necessary to identify the causal reason. Otherwise, please remove this sentence.
13. Line 266. If median is indicated, it should be also indicated IQR.
14. Line 266-267 Maybe a correlation test will help to support this result.
15. Line 270-271. This seems more appropriate to methods section, elucidating the reasons for sequencing some isolates with V14 chemistry.
16. Line 275. Regarding serotypes 15B and C, some common serological methods cannot differentiate 15B and 15C serotypes and interconversion events have also been described to occur between both serotypes. Consider any 15C or 15B result to be indicated as 15B/C and, therefore, a concordant result.
17. Line 281-282. Concordance between Illumina and WGS is not 100%, is 24/27. Or at least should be differently explained... (i.e among isolates with GPSC assigned by both ONT and Illumina (n=24), concordance was 100%...).

18. Line 283. The concordance between both approaches in AMR results against Sulfamethoxazole/trimethoprim was 93% (25/27). Concordance calculation cannot be clearly understood. What it may be assumed from the text: concordance for resistant isolates is obtained when COT resistance is determined by AST and any genetic resistance determinant is found for Trimethoprim or Sulfamethoxazole, but not necessary for both. In this case, the manuscript should benefit from including Trimethoprim-Sulfamethoxazole genetic resistance results as a combination, to better clarification and correlation between results and what is depicted in table 2.
19. Line 331. A threshold of 100x seems arbitrary. ¿Why to use 100x sequencing depth and not 80x?
20. Line 326. Could you please indicate Standard Deviation? I suggest to include time to perform the assembly in supplementary table S7 and S8, since there is an important conclusion derived from this data.
21. Line 335. Regarding the sentence: " Unicycler performed better than Pilon..." This is a conclusion to be placed on discussion. Conclusions and interpretations are mixed with results throughout the manuscript and would be better placed on Discussion section.
22. Line 344-347. This manuscript is more centered in the comparison of sequencing technologies and bioinformatic pipelines for the genomic characterization of *S. pneumoniae*. Due to the great amount of data, the manuscript should benefit from reducing the text and adding this Figure to supplementary files.
23. Line 371. Some antibiotics are not mentioned before: Kanamycin, linezolid...
24. Line 396-399. To reach such conclusion I have not seen any previous result. Please remove this sentence.
25. Line 389-392. This sentence is a result.
26. Figure 1C-1D. Median statistics may better reflect ONT data. Less than 1 kb of average length seems a short ONT read length.
27. Table 1. There are some points to consider in table 1. There are 6/27 isolates that are NT in the conventional multiplex PCR, but serogroup 24 and 19 by WGS. According to the reference indicated in the manuscript, primers are included in conventional PCR for serogroup 24 and 19. Why it is considered that such results are concordant: " The serogroups of 24/27 (89%) isolates predicted by WGS were concordant with multiplex PCR..." ? They are discrepant results, or at least results not to be considered.
28. Table 1. Another important point is that there is not enough evidence to discriminate serotypes from serogroup 24 at the genetic level. Therefore, the manuscript should benefit from informing PneumoKity results as serogroup 24. Further validation for differentiating specific serotypes is needed as the developers of PneumoKity claim itself. Therefore, these results regarding serogroup 24 are concordant with all WGS pipelines.
29. Table 1. Regarding S650 conventional result 8F. It is a typo?
30. Table 1. Regarding 19A-II in S685 is a subtype from serotype 19A, since the manuscript just refers to serotypes, just 19A serotype should be indicated.
31. Table 2. For a better understanding and rapid comparison of resistance profiles, please highlight in some way the resistances observed, more than bold.
32. Table 2. Please, add COT abbreviation.
33. Table 2. Treatment of pneumococcal infections usually include beta-lactam antibiotics and fluoroquinolones. ¿Why to include such antibiotics in table 2 and not beta-lactam or fluoroquinolones?
34. Table 2. Why to include Trimethoprim and Sulfamethoxazole genetic resistance per separate? The manuscript should benefit from including both as a combination, since they are only administered to patients in a combined way.

Reviewer #5 (Comments for the Author):

In this paper authors compare the Illumina and ONT systems for *S. pneumoniae* sequencing (27 isolates) and evaluate their performance for serotyping, MLST and antimicrobial resistance predictions.

I find this manuscript interesting to read and the results are certainly valuable to the research community. But many improvements are needed.

Serotyping failed for 6 out of 27 isolates by Multiplex PCR (labeled as NT). For these NT isolates, authors should use the gold standard Quellung reaction for typing or repeated PCR to resolve serotypes for those. Otherwise, you can't assume WGS typing results for those "NT" isolates are correct (lack of reference results).

The calculation in this sentence seems wrong, "The serogroups of 24/27 (89 %) isolates predicted by WGS were concordant with multiplex PCR (Table 1)."; I believe only 21 out of 27 (77.8%) were concordant (6 as NT by PCR). Please either perform confirmatory typing tests for NT isolates or correct the % concordance.

How many samples were barcoded and multiplexed, loaded on a single flow cell? Is it one sample per flow cell? Averaging how many reads for each sample? You should specify this in the Methods (or include a supp table).

Roughly how much cost per sample (including reagent and flow cell)? How much is the cost per sample for Illumina? Authors claim ONT has much lower cost, but data support is necessary (please include a comparison table).

It looks like ONT is worse for MLST (SNP typing) compared to Illumina due to higher error rates. I believe ONT has strength in resolving recombination events such as AMR gene cassettes, and resistant plasmid transfers. Have authors found evidence that ONT is better at this (any data)?

I think the main advantages of nanopore WGS versus Illumina in clinical settings is its rapid turn-around time due to its real-time sequencing and data analysis features that enable results (typing, AMR, outbreak tracking) within hours, instead of weeks for traditional methods. Do authors plan to streamline the pipeline for real-time prediction? Authors should include this in the discussion.

Minor comments:

Line 130-32, what does "The lack of need for sample batching" mean? Does it imply that authors suggest running 1 sample per flow cell?

In this manuscript, the authors perform an extensive and useful comparison of different second and third-generation sequencing technologies as well as different bioinformatic pipelines for pneumococcal genome assembly and its genomic characterization. The manuscript is of high interest to pneumococcal surveillance programs, in the light of the actual context of switching from classic phenotypic and molecular methods to whole genome sequencing methods. The sequencing chemistry and bioinformatic programs are continuously improving to better reflect the real biological sequences of the pathogens studied, and literature has to reflect these changes. This manuscript is another evidence that supports the use of ONT sequencing for its implementation in the rapid *S. pneumoniae* genomic characterization in a cost-effective and flexible way, without the need of important computer resources. However, the latter should be better reflected on the manuscript since it is an important conclusion. In addition, specific attention should be paid throughout the manuscript to avoid mixing results with interpretation and conclusions. Finally, there is a great amount of data and the manuscript should benefit from a more concise presentation. Specific questions to address are found below:

1. Line 77. Suggest to replace ONT kit v14 by ONT_V14
2. Line 87 and 88. Suggest to replace the **most** in-depth by **an** in-depth.
3. Line 76. Typo in *S. pneumoniae*
4. Line 95. Phenotypic testing is not performed in this paper, so this affirmation does not seem correct on the importance section.
5. Line 123-124, line 136. Please, replace diagnostic by typing. In addition, all disadvantages do not apply to all conventional typing methods. Please, this sentence should be reformulated.
6. Line 126. Regarding this affirmation: Illumina entails high costs of sample processing... Illumina is one of the most competitive sequencing platforms in terms of accuracy and price. The problem relies on the need to accumulate great number of samples to achieve such low cost, which increases in time-to-results. This sentence should be clarified.
7. Line 176. ¿Why beta-lactam antibiotics were not tested? In Table S5 there are indicated AST results from penicillin.
8. Line 202-242 Why to compare specifically porechop/canu and assembleBAC-ONT and not others. ¿Which is the interest? ¿Are there differences that make them interesting to compare? The manuscript should benefit from a brief explanation.
9. Line 201. There is a high level of confusion on this section. There are results such as MLST, resistance genes, serotype, etc. obtained by both the gps-unified pipeline and Pathogenwatch, but only the latter from Pathogenwatch are considered. This section should be simplified. The manuscript should benefit from just mention the pipelines and tools used within these pipelines until the obtention of the assembled genome, which is the input to Pathogenwatch.
10. Line 251-253. This is very theoretic and could be added to supplementary material.
11. Line 255-259. This is a mix of interpretation and discussion, therefore it should be better placed in other section
12. Line 263. The manuscript mentions: “canu may have generated over-assembled genomes containing an overrepresentation of repeated region”. This sentence is more appropriate in the discussion section. Moreover, the interpretation may be not supported with the data provided. There are several reasons that can impact genome contiguity, such as the choice of parameters of minimum overlap length or k-mer size, the way they are handling repetitive regions, different abilities to correct sequencing errors (sequencing errors can be interpreted as new real biological sequences), how to treat genome regions with low coverage (some assemblers may break contigs in areas with low coverage to avoid uncertainty, leading to more contigs), and

several additional reasons. Additional analyses will be necessary to identify the causal reason. Otherwise, please remove this sentence.

13. Line 266. If median is indicated, it should be also indicated IQR.

14. Line 266-267 Maybe a correlation test will help to support this result.

15. Line 270-271. This seems more appropriate to methods section, elucidating the reasons for sequencing some isolates with V14 chemistry.

16. Line 275. Regarding serotypes 15B and C, some common serological methods cannot differentiate 15B and 15C serotypes and interconversion events have also been described to occur between both serotypes. Consider any 15C or 15B result to be indicated as 15B/C and, therefore, a concordant result.

17. Line 281-282. Concordance between Illumina and WGS is not 100%, is 24/27. Or at least should be differently explained... (i.e among isolates with GPSC assigned by both ONT and Illumina (n=24), concordance was 100%...).

18. Line 283. The concordance between both approaches in AMR results against Sulfamethoxazole/trimethoprim was 93% (25/27). Concordance calculation cannot be clearly understood. What it may be assumed from the text: concordance for resistant isolates is obtained when COT resistance is determined by AST and any genetic resistance determinant is found for Trimethoprim or Sulfamethoxazole, but not necessary for both. In this case, the manuscript should benefit from including Trimethoprim-Sulfamethoxazole genetic resistance results as a combination, to better clarification and correlation between results and what is depicted in table 2.

19. Line 331. A threshold of 100x seems arbitrary. ¿Why to use 100x sequencing depth and not 80x?

20. Line 326. Could you please indicate Standard Deviation? I suggest to include time to perform the assembly in supplementary table S7 and S8, since there is an important conclusion derived from this data.

21. Line 335. Regarding the sentence: “Unicycler performed better than Pilon...” This is a conclusion to be placed on discussion. Conclusions and interpretations are mixed with results throughout the manuscript and would be better placed on Discussion section.

22. Line 344-347. This manuscript is more centered in the comparison of sequencing technologies and bioinformatic pipelines for the genomic characterization of *S. pneumoniae*. Due to the great amount of data, the manuscript should benefit from reducing the text and adding this Figure to supplementary files.

23. Line 371. Some antibiotics are not mentioned before: Kanamycin, linezolid...

24. Line 396-399. To reach such conclusion I have not seen any previous result. Please remove this sentence.

25. Line 389-392. This sentence is a result.

26. Figure 1C-1D. Median statistics may better reflect ONT data. Less than 1 kb of average length seems a short ONT read length.

27. Table 1. There are some points to consider in table 1. There are 6/27 isolates that are NT in the conventional multiplex PCR, but serogroup 24 and 19 by WGS. According to the reference indicated in the manuscript, primers are included in conventional PCR for serogroup 24 and 19. Why it is considered that such results are concordant: “The serogroups of 24/27 (89%) isolates predicted by WGS were concordant with multiplex PCR...”? They are discrepant results, or at least results not to be considered.

28. Table 1. Another important point is that there is not enough evidence to discriminate serotypes from serogroup 24 at the genetic level. Therefore, the manuscript should benefit from informing PneumoKity results as serogroup 24. Further validation for differentiating specific serotypes is needed as the developers

of PneumoKity claim itself. Therefore, these results regarding serogroup 24 are concordant with all WGS pipelines.

29. Table 1. Regarding S650 conventional result 8F. It is a typo?

30. Table 1. Regarding 19A-II in S685 is a subtype from serotype 19A, since the manuscript just refers to serotypes, just 19A serotype should be indicated.

31. Table 2. For a better understanding and rapid comparison of resistance profiles, please highlight in some way the resistances observed, more than bold.

32. Table 2. Please, add COT abbreviation.

33. Table 2. Treatment of pneumococcal infections usually include beta-lactam antibiotics and fluoroquinolones. ¿Why to include such antibiotics in table 2 and not beta-lactam or fluoroquinolones?

34. Table 2. Why to include Trimethoprim and Sulfamethoxazole genetic resistance per separate? The manuscript should benefit from including both as a combination, since they are only administered to patients in a combined way.

Dear Dr. Sophia Georghiou,

We would like to submit our revised manuscript “Comparison of Illumina and Oxford Nanopore Technology systems for the genomic characterization of *Streptococcus pneumoniae*”.

We appreciate the insightful comments made by the editor and reviewers. We thank them for the time and effort spent reviewing the manuscript.

We have carefully read the comments from the second revision round and responded to them individually, indicating how we addressed each comment and describing the changes amended.

The revisions have been approved by all authors. The changes are marked in tracking mode in the manuscript. Our responses to the referee’s comments are included below.

We hope that the revisions will meet the reviewers’ approval.

Reviewer #4 (Public repository details (Required)):

Genome assemblies or raw fastq data should be deposited in a public repository such as ENA, SRA.

Response: The assemblies and annotations for all *S. pneumoniae* isolates included in this study were uploaded and made public on Pathogenwatch, which constitutes a large repository of curated bacterial genomic data.

The “Data Availability section” of the paper includes a link that provides access to these files:

Page 19, Lines 440-444: {All supporting data and protocols are provided within the article or through supplemental material. Pneumococcal genomes from hybrid assembly and their annotations were uploaded to Pathogenwatch. They can be accessed and downloaded using the following link: <https://pathogen.watch/collection/ui40ojhh5yyw-hybrid-assembly>}

Reviewer #4 (Comments for the Author):

*In this manuscript, the authors perform an extensive and useful comparison of different second and third-generation sequencing technologies as well as different bioinformatic pipelines for pneumococcal genome assembly and its genomic characterization. The manuscript is of high interest to pneumococcal surveillance programs, in the light of the actual context of switching from classic phenotypic and molecular methods to whole genome sequencing methods. The sequencing chemistry and bioinformatic programs are continuously improving to better reflect the real biological sequences of the pathogens studied, and literature has to reflect these changes. This manuscript is another evidence that supports the use of ONT sequencing for its implementation in the rapid *S. pneumoniae* genomic characterization in a cost-effective and flexible way, without the need of important computer resources. However, the latter should be better reflected on the manuscript since it is an important conclusion. In addition, specific attention should be paid throughout the manuscript to avoid mixing results with interpretation and conclusions. Finally, there is a*

great amount of data and the manuscript should benefit from a more concise presentation. Specific questions to address are found below:

Response: Thank you for this positive evaluation and for the insightful and constructive feedback, which contributed to enhancing the quality of our manuscript. We have made several revisions to address all concerns and suggestions raised. The manuscript was reviewed to ensure that the presentation of results is distinct from their interpretation and conclusions. Moreover, we have refined the text to present our findings more concisely, without compromising the impact of the findings.

We appreciate your comment on emphasizing the benefits of ONT sequencing. This was already included in the last paragraph of the “Discussion” section. The latter explicitly highlights the cost-effectiveness, flexibility, and minimal computational resources required by ONT sequencing. Moreover, we have incorporated evidence from literature regarding the promising ability of the newer ONT chemistry to enhance sequencing accuracy and performance, further supporting its suitability for rapid *S. pneumoniae* genomic characterization. The mentioned paragraph is included below:

Page 18, Lines 415-435: {We demonstrated the reliability of the ONT V10 kit and R9.4.1 flow cells for *S. pneumoniae* identification, serotyping, GPSC and AMR prediction. Implementing ONT sequencing in clinical settings is feasible due to its low costs, the improved accuracy by the V14 chemistry and the high concordance between conventional diagnostics results and WGS data. The flexibility, portability and reproducibility of the proposed pipelines further facilitate the implementation of the technique in research and clinical laboratories. Accordingly, this technique paves the way for LMICs to harness the benefits of genomics-led health care. ONT and downstream analysis are continuously improving, which may facilitate high-resolution genotyping from ONT-only data in the future (13). A recent study by Hong et al. revealed that ONT data generated by R10.4.1 flow cells exhibited accuracy that is comparable to Illumina sequences (34). The authors used the Dorado 0.5.0 Super Accurate 4.3 basecalling model, which is more accurate than the model used in this study. The utilization of super-accuracy models in this study was limited by the availability of computing and time resources. Another study by Bogaerts et al. revealed that quality scores were improved in R10 compared to R9 data in *E. coli* and *L. monocytogenes* (33). Moreover, the authors constructed SNP-based *E. coli* phylogenies from ONT R10 or Illumina data. They demonstrated that SNP distance matrices and tree topologies were almost identical between both technologies. Another study demonstrated that genomes assembled with ONT-only data contained less than 100 single nucleotide variants (SNVs) compared to hybrid assemblies (29). The capacity of ONT_V14 to improve AMR prediction, MLST and pbp typing, as demonstrated in this study, further highlights the promising potential of Nanopore technology for comprehensive bacterial sequencing and analysis.}

1. Line 77. Suggest to replace ONT kit v14 by ONT_V14

Response: The replacement was made as suggested:

Page 3, Line 77: {The ONT_V14 chemistry significantly improved both MLST and pbp prediction in long-read sequencing.}

2. Line 87 and 88. Suggest to replace the most in-depth by an in-depth.

Response: Thank you for this suggestion. The sentence was modified as follows:

Page 4, Lines 87-89: {It represents an in-depth investigation of Illumina and Oxford Nanopore technologies (ONT) systems for bacterial sequencing.}

3. Line 76. Typo in *S. pneumonia*

Response: The typo was corrected as follows:

Page 3, Lines 76-77: {*S. pneumoniae* identification, serotyping, AMR and GPSC prediction were successfully achieved using ONT sequencing.}

4. Line 95. Phenotypic testing is not performed in this paper, so this affirmation does not seem correct on the importance section.

Response: AMR was assessed by a phenotypic detection method. Particularly, antimicrobial susceptibility testing (AST) was performed using disk diffusion and the E-test to determine resistance profiles against several antibiotics.

Serotyping, on the other hand, was performed using a classic molecular method (multiplex PCR).

Hence, the sentence was modified as follows:

Page 4, Lines 93-96: {We report a strong case for the implementation of the WGS technique in the clinical setting based on its high concordance with conventional molecular and phenotypic methods. Furthermore, the flexibility and portability of the investigated pipelines facilitates their use in clinical applications.}

5. Line 123-124, line 136. Please, replace diagnostic by typing. In addition, all disadvantages do not apply to all conventional typing methods. Please, this sentence should be reformulated.

Response: The term “diagnostic” was replaced with “typing” as suggested.

Moreover, the sentence on the disadvantages of conventional typing methods was attenuated as follows:

Page 5, Lines 122-124: {WGS-based approaches for pneumococcal characterization represent an attractive alternative to conventional typing methods, which can be time-consuming and labor-intensive (10, 11).}

This reformulated sentence does not imply that all conventional methods suffer from the previously mentioned pitfalls. Moreover, references 10 and 11 included at the end of the sentence contain comprehensive reviews of AST and capsular typing methods, including both their advantages and disadvantages.

6. **Line 126. Regarding this affirmation: Illumina entails high costs of sample processing... Illumina is one of the most competitive sequencing platforms in terms of accuracy and price. The problem relies on the need to accumulate great number of samples to achieve such low cost, which increases in time-to-results. This sentence should be clarified.**

Response: Indeed, Illumina sequencing is more expensive compared to ONT, especially considering the prices of the sequencing devices. To demonstrate this, we have prepared a table that compares the sequencing cost of 90 *S. pneumoniae* isolates using both platforms. The table is solely based on the prices of sequencing kits and flow cells.

First, the number of Illumina flow cells needed for the sequencing of 90 isolates is calculated based on the following parameters:

- Genome size= 2,200,000 bp
- MiSeq output= 7 Gb
- Target depth= 100x

Based on the above parameters, the number of samples to multiplex is ~32 isolates (formula included in the below illustration). At our lab, the maximum number of *S. pneumoniae* samples allowed in a MiSeq flow cell is 30.

Thus, we need 3 Illumina Miseq flow cells to sequence a total of 90 *S. pneumoniae* isolates.

Library sample multiplexing

$$\text{\#sample to multiplex} = \frac{\text{total sequencer output}}{(\text{min. depth coverage} * \text{total genome target})}$$

The total combined genome size of the bacterial samples loaded into an ONT flow cell should not exceed 35 million base pairs (bp). This limit was established through previous in-house experiments, which concluded that exceeding this combined genome size can negatively impact the total reads output.

Thus, each ONT flow cell can accommodate a total of 15 *S. pneumoniae* isolates (15 x 2,200,000 bp = 33,000,000 bp, which is lower than the 35 million bp cutoff).

Hence, a total of 6 ONT flow cells are needed for the sequencing of 90 *S. pneumoniae* isolates.

The below table shows that ONT (5,190\$) cuts the sequencing cost of 90 *S. pneumoniae* isolates almost by half compared to Illumina (9,930\$):

	Cat#	Reagent		Price/kit	Total
Illumina*	ILM20060059	Illumina DNA Prep, (M) Tagmentation (96 Samples, IPB)	1	4,010 \$	4,010 \$
	ILM20091654	Illumina DNA/RNA UD Indexes Set A, Tagmentation (96 Indexes, 96 Samples)	1	610 \$	610 \$
	ILM15033411	MiSeq Reagent Kit v2 (500-cycles)	3	1,770 \$	5,310 \$
	Grand Total (90 samples)				
ONT	SQK-RBK114.96	Rapid Barcoding Kit 96 V14	1	990	990
	FLO-MIN114	Flow Cell (R10.4.1)	6	700	4,200 \$
	Grand Total (90 samples)				

*Illumina prices are provided by suppliers in Lebanon. Like many LMICs, local suppliers in Lebanon are resellers who charge large premiums.

A study by Boostrom et al. included a cost analysis of both platforms based on UK prices [1]. They revealed that Illumina does entail higher costs of sample processing compared to ONT. The authors calculated start-up costs for equipment and consumables capable of generating 48 isolates. The start-up cost was estimated to be £ 107,117 for Illumina and £ 11,369 for ONT/Mk1c.

The Illumina NextSeq system is notably more cost-effective than MiSeq and ONT when only considering the costs of sequencing kits and flow cells. However, MiSeq or ONT are better suited for sequencing smaller batches of samples without significant waste. In contrast, the NextSeq workflow is optimized for high-throughput sequencing. Hence, clinical labs processing fewer isolates may face delays in sequencing while waiting to accumulate enough samples for a NextSeq run.

7. Line 176. ¿Why beta-lactam antibiotics were not tested? In Table S5 there are indicated AST results from penicillin.

Response: As mentioned by the reviewer, AST and WGS-based penicillin AMR data, as well as predicted pbp types, are included in supplementary tables S5 and S6.

Penicillin AST lab methodology was added to the “Materials and Methods” section as follows:

Page, Line: {Antimicrobial susceptibility testing (AST) was performed using disk diffusion to determine resistance profiles against chloramphenicol, clindamycin, erythromycin, sulfamethoxazole/trimethoprim, tetracycline and levofloxacin and the E-test (Biomerieux, France) for penicillin following the CLSI guidelines (19). Penicillin minimum inhibitory concentration (MIC) was interpreted based on the meningitis MIC cut-off. Penicillin with MIC>0.06 µg/ml was categorized as resistant.}

Of note, AST and WGS data is also available for ceftriaxone, a key beta lactam antibiotic. We opted to include as much information as possible on AMR against several antibiotics, while aiming for conciseness. Thus, we preferred not to add a table on ceftriaxone to the supplementary materials. Instead, ceftriaxone AMR concordance between the sequencing techniques was mentioned briefly in the manuscript as follows:

Page 13, Lines 297-299: {Furthermore, the concordance between Illumina and ONT_V10 was 100% for ceftriaxone, kanamycin and linezolid (data not shown).}

8. Line 202-242 Why to compare specifically porechop/canu and assembleBAC-ONT and not others. ¿Which is the interest? ¿Are there differences that make them interesting to compare? The manuscript should benefit from a brief explanation.

Response: While comparing several assemblers could be valuable, it was not the primary focus of this study. The Canu tool had been previously used in our lab for analyzing long-read data before we transitioned to the Flye-based pipeline. Upon observing the improved quality of Flye assemblies, we decided to report this finding for the benefit of other researchers considering similar approaches.

9. Line 201. There is a high level of confusion on this section. There are results such as MLST, resistance genes, serotype, etc. obtained by both the gps-unified pipeline and Pathogenwatch, but only the latter from Pathogenwatch are considered. This section should be simplified. The manuscript should benefit from just mention the pipelines and tools used within these pipelines until the obtention of the assembled genome, which is the input to Pathogenwatch.

Response: We appreciate this comment, which improved the conciseness of this section.

The “Data Analysis” subsection in the “Materials and Methods” section was modified as follows:

Page 9, Lines 207-217: {Illumina data were analyzed using the gps-unified pipeline (<https://github.com/sanger-bentley-group/gps-pipeline/tree/v1.0.0-rc2>) which utilizes SPAdes (v1.1.0) for assembly and QUAST (v5.0.2) for assembly QC.

FastQ files from long-read sequencing were analyzed with the assemble-BAC-ONT pipeline (<https://github.com/avantonder/assembleBAC-ONT>) version 1.1 to generate polished FASTA files using Flye (v2.9-b1768) and medaka (v1.4.4). The pipeline additionally generates QC reports using MultiQC and checkm2 (v1.0.1).

FASTA files obtained from both pipelines were analyzed using Pathogenwatch (<https://pathogen.watch/>) to obtain serotyping, AMR, MLST and Global Pneumococcal Sequence Clusters (GPSC) predictions.}

The reason why additional outputs from the gps-unified pipeline were mentioned initially was to inform readers that it can be utilized without the need to upload FASTA files to Pathogenwatch. We utilized Pathogenwatch to maximize uniformity across results from ONT and Illumina. The gps-unified pipeline users will not have to do so, because the pipeline generates serotyping, MLST, AMR, virulence and GPSC data.

10. Line 251-253. This is very theoretic and could be added to supplementary material.

Response: Subsection 3.1 in the “Results” section was reformulated to remove theoretic notions or interpretations as follows:

Page 11, Lines 247-254: {The fastq-scan tool was used for raw data quality assessment. The median coverage was 58.8X and 77.29X for ONT_V10 and Illumina sequences respectively (Figure 1A). MiSeq sequencing generated substantially more reads than ONT_V10 (Figure 1B). However, the average read length in ONT_V10 was higher than that of Illumina (Figure 1C). The maximum read length was impressively high in ONT_V10, ranging from 17,469 bp to 288,361 bp (Table S2).

Figure 1D shows that the average quality score of Illumina sequences (n=27; Mean, 34.37; SD, 0.88) was higher than that of ONT_V10 (n=27; Mean, 18.37; SD, 0.68). These quality scores correspond to Phred scores, which quantify the error probability in a base call.}

11. Line 255-259. This is a mix of interpretation and discussion, therefore it should be better placed in other section

Response: The interpretation was removed from the “Results” section and incorporated into the “Discussion” section as follows:

Page 17, Lines 384-392: {Most isolates processed by ONT_V10 had novel STs due to missing data in MLST alleles. This phenomenon may be attributed to inaccurate assemblies within sequence typing regions, which are sensitive to variations in SNPs. Downstream analyses, such as variant calling and genome assembly accuracy may have been affected by the reduced mean quality score initially obtained in the ONT_V10 reads. ONT_V14 improved MLST typing in 11/12 isolates. This may be attributed to the much lower error rates expected from the V14 kits and R10 flow cells. Bogaerts et al. showed that the data generated from ONT R10 flow cells had very similar trends to those from Illumina, while ONT R9 data deviated slightly from it (33). The authors additionally found that the newer ONT chemistry could generate accurate and reliable SNP-based phylogenies.}

12. Line 263. The manuscript mentions: "canu may have generated over-assembled genomes containing an overrepresentation of repeated region". This sentence is more appropriate in the discussion section. Moreover, the interpretation may be not supported with the data provided. There are several reasons that can impact genome contiguity, such as the choice of parameters of minimum overlap length or k-mer size, the way they are handling repetitive regions, different abilities to correct sequencing errors (sequencing errors can be interpreted as new real biological sequences), how to treat genome regions with low coverage (some assemblers may break contigs in areas with low coverage to avoid uncertainty, leading to more contigs), and several additional reasons. Additional analyses will be necessary to identify the causal reason. Otherwise, please remove this sentence.

Response: Indeed, the mentioned sentence is an interpretation that is now removed from the “Results” section:

{Canu may have generated over-assembled genomes containing an overrepresentation of repeated regions}

Of note, the above sentence represented a possible explanation of why Canu assemblies had a higher genome size (bp) in all isolates compared to those from the assembleBAC-ONT. It was not intended to explain the differences in genome contiguity. The genome sizes obtained from the pipeline were closer to those obtained by Illumina in all isolates.

We did find that the pipeline enhanced contiguity compared to Canu, but did not attempt to explain the result as mentioned in the comment. Regarding Canu parameters, the minReadLength (Default: 1000) and minOverlapLength (Default: 500) options were not modified and maintained at their default settings. Hence, other factors mentioned in the comment could have potentially accounted for the observed reduction in contiguity. However, we did not conduct analyses to identify these factors.

Similar to our findings, a study by Wick et al. compared eight long-read assemblers for bacterial WGS analysis including, Canu, NECAT, Raven, Shasta, Redbean, NextDenovo/NextPolish, Flye, Miniasm/Minipolish and Flye [2]. The authors found that Canu performed poorly with circularization among all assemblers tested. Boostrom et al. also compared bacterial genomic assemblies from Canu, Flye, Raven and Miniasm, revealing that Canu produced the most fragmented assemblies [1].

13. Line 266. If median is indicated, it should be also indicated IQR.

Response: The sentence was amended to include the IQR as follows:

Page 12, Lines 264-265: {The median contig number was 20 in ONT_V10 (IQR=79) and 40 in Illumina (IQR=20) (**Figure 1E**).}

14. Line 266-267 Maybe a correlation test will help to support this result.

Response: We assessed the covariation between the ONT sequencing depth and contiguity (X-Y plot in Figure 1G) by computing the Pearson’s correlation coefficient (r) in GraphPad Prism. A significant negative correlation between the number of contigs and the sequencing depth was obtained as follows:

Pearson r	
r	-0.4683
95% confidence interval	-0.7202 to -0.1074
R squared	0.2193
P value	
P (two-tailed)	0.0137
P value summary	*
Significant? (alpha = 0.05)	Yes
Number of XY Pairs	27

Modifications added to the manuscript:

Figure 1G was removed, since it depicts the same idea as Figure 1F (higher sequencing depth is associated with a lower number of contigs). This change was made to avoid repetition and to render the figure cleaner and the manuscript more concise.

To support the result mentioned in the comment, we included statistical analysis of the groups in Figure 1F, which are categorized by sequencing depth as follows:

- Group 1: Isolates with a sequencing depth <50X
- Group 2: Isolates with a sequencing depth > 50X, but below 100X
- Group 3: Isolates with a sequencing depth >100X

Of note, group 2 had previously included all isolates with sequencing depth > 50X. Now it does not contain isolates with depth >100X.

Figure 1F now includes statistical associations as follows:

Accordingly, the legend of Figure 1 was modified to include interpretations of the p-values in Figure 1F:

Page 26, Line 589: {Significant values are indicated with **($p < 0.01$) or ***($p < 0.001$) (Mann-Whitney test)}

Moreover, statistical analysis was added to the “Materials and Methods” section as follows:

Page 11, Lines 242-244: {GraphPad for Windows (v8.4) (GraphPad Software, California, USA) was used for statistical analysis. Normality was assessed using the Kolmogorov–Smirnov test and mean quality scores were compared using the Mann-Whitney test.}

Finally, we added a brief description of the results in Figure 1F into the “Results” section as follows:

Page 12, Lines 265-268: {The number of contigs ranged between 1 and 5 in ONT_V10 samples with sequencing depth > 100X (Figure 1F). Isolates with depth > 100X had significantly lower contigs compared to those with depths of 50X to 100X ($p = 0.002$) and to isolates with depth < 50X ($p < 0.0001$).}

15. Line 270-271. This seems more appropriate to methods section, elucidating the reasons for sequencing some isolates with V14 chemistry.

Response: Based on your comment the following sentence was removed from the “Results” section:

{Samples with abnormal genome size or guanine-cytosine content (GC-content) in ONT_V10 were sequenced by ONT_V14 (data not shown).}

Moreover, the “Materials and Methods” section now includes an explanation of how samples were selected for sequencing using the new ONT chemistry, by referring to Figure S1:

Page 9, Lines 201-203: {Furthermore, a total of 12 isolates were selected for sequencing using the most recent kit 14 chemistry and R10.4.1 flow cells from ONT (ONT_V14). The rationale for selecting these isolates is detailed in Figure S1.}

16. Line 275. Regarding serotypes 15B and C, some common serological methods cannot differentiate 15B and 15C serotypes and interconversion events have also been described to occur between both serotypes. Consider any 15C or 15B result to be indicated as 15B/C and, therefore, a concordant result.

Response: The serotypes included in Table 1 represent the exact output from Pathogenwatch. For example, Pathogenwatch predicted the serotype as 15C in S670 based on ONT and Illumina data, with no mention of serotype 15B. While we agree with your point, we opted to present the data as it was output by Pathogenwatch to ensure transparency and accuracy in reporting the tool’s predictions. Hence, the isolates were not assigned as 15B/C in Table 1.

We did, however, remove the following sentences from the manuscript:

- {However, three samples that were assigned to serogroup 15 were variable at the serotype level.}
- {Moreover, PneumoKITY predicted the serotypes of S682 and S683 obtained from ONT_V10 as either 15B or 15C, both exhibiting the same maximum k-mer percentages.}

17. Line 281-282. Concordance between Illumina and WGS is not 100%, is 24/27. Or at least should be differently explained... (i.e among isolates with GPSC assigned by both ONT and Illumina (n=24), concordance was 100%...).

Response: Thank you for this comment. The sentence was amended as suggested:

Page 12, Lines 277-278: {Among isolates with GPSC assigned by both ONT and Illumina (n=24), the concordance in predicted GPSC was 100%.}

18. Line 283. The concordance between both approaches in AMR results against Sulfamethoxazole/trimethoprim was 93% (25/27). Concordance calculation cannot be clearly understood. What it may be assumed from the text: concordance for resistant isolates is obtained when COT resistance is determined by AST and any genetic resistance determinant is found for Trimethoprim or Sulfamethoxazole, but not necessary for both. In this case, the manuscript should benefit from including Trimethoprim-Sulfamethoxazole genetic resistance

results as a combination, to better clarification and correlation between results and what is depicted in table 2.

Response: Based on your recommendation, Trimethoprim and Sulfamethoxazole genetic resistance is now included as a combination in Table 2. Indeed, the below notion in the comment is accurate:

“Concordance for resistant isolates is obtained when COT resistance is determined by AST and any genetic resistance determinant is found for Trimethoprim or Sulfamethoxazole, but not necessary for both”

19. Line 331. A threshold of 100x seems arbitrary. ¿Why to use 100x sequencing depth and not 80x?

Response: The threshold was chosen based on the results presented in Table S7, which includes all isolates at a depth > 100X (n=8). Among these, Pilon enhanced contiguity in 4/8 (50 %) isolates and improved the N50 parameter in 2/8 (25 %). In the remaining 4/8 (50 %) isolates, the QC parameters were almost identical with Unicycler. Hence the recommendation of Pilon utilization at ONT depth > 100X.

The sequencing depth of the remaining isolates in the study (n=19) ranged between 37X and 76X. Hence, it was not possible to explore hybrid assembly approaches in isolates with depths ranging between 80X and 100X.

Based on our data, neither Pilon nor Unicycler consistently outperformed the other in isolates with sequencing depths between 50X and 76X. Unicycler improved the N50 and contig number parameters in only 7/10 of these isolates (70 %). To illustrate this variability, let's examine two isolates with similar depths in the mentioned range: S674 (72X) and S685 (71X).

In S685, Pilon generated a higher number of contigs compared to Unicycler. The case was opposite for S674, in which less contigs were generated by Pilon.

The following sentence was amended to reflect more accurately the findings of hybrid assembly analysis in this study:

Page 15, Lines 334-336: {At ONT depths between 50X and 76X (n=10), Unicycler improved the N50 and contig number parameters in 7/10 isolates (70 %).}

20. Line 326. Could you please indicate Standard Deviation? I suggest to include time to perform the assembly in supplementary table S7 and S8, since there is an important conclusion derived from this data.

Response: Including the standard deviation (SD) would require precise data on the time taken for each tool to process each isolate. Currently, the reported durations are rough estimations based on observation rather than exact records. To calculate the SD, a new analysis would be needed, involving detailed tracking of the time each tool takes to complete each assembly. Although this request is reasonable, it may not be practical, as it would require repeating the analysis solely for time tracking purposes (which was not a main aim of our analysis). Many authors report in

literature that certain bioinformatics tools are more-time consuming or computationally demanding compared to others, without reporting the exact time in minutes or providing a screenshot of the CPU/RAM data usage when each tool is running.

The word “average” was removed from the sentence as follows:

Page, Line: {Notably, Pilon was less time-consuming, taking around 4 min/sample for hybrid assembly. Unicycler required ~60 min to process files with ONT depth >100X.}

21. Line 335. Regarding the sentence: " Unicycler performed better than Pilon..." This is a conclusion to be placed on discussion. Conclusions and interpretations are mixed with results throught the manuscript and would be better placed on Discussion section.

Response: We thank the reviewer for pointing out this issue and appreciate the opportunity to further refine the manuscript. The “Results” section was thoroughly reviewed to identify and remove conclusions and interpretations.

Moreover, the mentioned sentence was modified as follows:

Page 15, Lines 333-334: {At ONT depths< 50X (n=9), Unicycler generated higher N50 parameters and fewer contigs compared to Pilon (**Table S8**).}

22. Line 344-347. This manuscript is more centered in the comparison of sequencing technologies and bioinformatic pipelines for the genomic characterization of *S. pneumoniae*. Due to the great amount of data, the manuscript should benefit from reducing the text and adding this Figure to supplementary files.

Response: Thank you for this suggestion. However, we prefer to keep the phylogenetic tree, since it provides important information that may be of interest to the audience. The latter may include researchers focused on *S. pneumoniae* and public health officials interested in bacterial WGS. It is noteworthy that isolates of serogroup 24/ST15253 or serotype 15B exhibited identical AMR predictions, inferring simultaneous resistance to penicillin, clindamycin, erythromycin, tetracycline, and sulfamethoxazole. The emergence of MDR serotype 15B and serogroup 24 causing IPD recently in Lebanon underscores important findings obtained from WGS.

Less significant information, like Type 1 pili expression, was removed from the text.

23. Line 371. Some antibiotics are not mentioned before: Kanamycin, linezolid...

Response: Thank you for this comment. Indeed, we did not include the WGS-based AMR results of these antibiotics in the manuscript to maintain clarity and avoid making the tables overly dense.

We removed the concordance summary of ceftriaxone, kanamycin and linezolid from the “Discussion” section and added them to the “Results” section. We indicated that their results are not presented in the manuscript:

Page 13, Lines 297-299: {Furthermore, the concordance between Illumina and ONT_V10 was 100% for ceftriaxone, kanamycin and linezolid (data not shown).}

24. Line 396-399. To reach such conclusion I have not seen any previous result. Please remove this sentence.

Response: The following sentence was removed because the data is not presented in the manuscript:

{The pipeline’s results matched those from Pathogenwatch in all isolates}

We believe it is important to discuss our experience with the pipeline, which fits well in the “Discussion” section. Hence, the below sentence was retained:

{However, the gps-unified pipeline provides a reliable, fast and practical workflow for Illumina-only data analysis. It outputs AMR, serotyping, MLST, GPSC, virulence and QC data.}

It intends to provide information on a practical pipeline that can be utilized by other researchers in the field or clinicians to simplify their analysis of Illumina-only data. Once they run the pipeline, there is no need for them to upload their FASTA files to pathogenwatch to obtain additional information:

We are only listing the pipeline outputs, which can be found at the developer’s GitHub page, referenced in the “Data analysis” subsection.

25. Line 389-392. This sentence is a result.

Response: Thank you for this comment. We moved the information on the time consumed by the assemble-Bac-ONT pipeline for assembly (based on the configuration needed to run it on our local PC) from the “Discussion” to the “Results” section as follows:

Page 11, Lines 256-263: {ONT_V10 genome assemblies were compared between porechop/canu and the assemble-BAC-ONT pipeline. The latter enhanced contiguity and GPSC identification compared to Canu (Table S3). Moreover, the pipeline generated assemblies with reduced genome size in 24/26 isolates. Canu took a long time —up to six hours in one isolate— to finalize the de novo assembly. The assemble-Bac-ONT pipeline was configured to utilize 11.GB max_memory, 6 max_cpus and 12.h max_time to overcome local computational limitations. Despite this, it was generally fast, taking around three hours to complete the analysis of 11 samples. We utilized the assemble-BAC-ONT pipeline as the standard approach for ONT analysis.}

26. Figure 1C-1D. Median statistics may better reflect ONT data. Less than 1 kb of average length seems a short ONT read length.

Response:

Figure 1C: We re-examined the fastq-scan outputs, and the median length was lower than the mean length in most isolates. This is due to the abundance of reads with short lengths in the raw fastq files, which the tool uses to generate the raw QC reports. The presence of these reads decreases both the median and average length of reads. However, all reads with short lengths are filtered out during assembly.

Figure 1D: The ONT_V10 mean quality score obtained in this study (18.37) is higher than what has been reported in two very recent studies from literature. The mean Q-score from the rapid barcoding kit v10 was reported to be below 14 in a study by Lerminiaux et al [3]. Similarly, Linde et al. revealed that 7–49% of bases on average reached Q15 with ONT sequencing using R9 flow cells [4].

27. Table 1. There are some points to consider in table 1. There are 6/27 isolates that are NT in the conventional multiplex PCR, but serogroup 24 and 19 by WGS. According to the reference indicated in the manuscript, primers are included in conventional PCR for serogroup 24 and 19. Why it is considered that such results are concordant: " The serogroups of 24/27 (89%) isolates predicted by WGS were concordant with multiplex PCR..." ? They are discrepant results, or at least results not to be considered.

Response: Based on the recommendation of reviewer 5, typing by multiplex PCR was repeated for the six mentioned isolates. Below are the results of the experiment:

Isolate	Serotype (PCR)
645	23A
647	18
664	NT
674	24A/24B/24F
676	24A/24B/24F
685	19A

S664 had an inconclusive result, showing only a very faint band at the serogroup 24 position on gel. Thus, we kept it assigned as non-typeable.

Table 1 was updated with the above results. Moreover, the serotyping results in the manuscript were updated as follows:

Page 12, Lines 270-272: {Sample S664 (1/27) was NT by PCR, and therefore, not included in the concordance analysis between PCR and WGS. The serogroups of 23/26 (88 %) isolates predicted by WGS were concordant with multiplex PCR (**Table 1**).}

28. Table 1. Another important point is that there is not enough evidence to discriminate serotypes from serogroup 24 at the genetic level. Therefore, the manuscript should benefit from informing PneumoKity results as serogroup 24. Further validation for differentiating specific serotypes is needed as the developers of PneumoKity claim itself. Therefore, these results regarding serogroup 24 are concordant with all WGS pipelines.

Indeed, genotypic approaches such as PneumoCat can only predict up to the serogroup level for certain serogroups such as 24 and 32 [5].

Pneumokity developers indicated that they have updated the Capsular Type Variant database (CTVdb), upon which PneumoCat also relies, to improve serotype detection accuracy [6]. This was done particularly for the challenging serotypes 24F, 33F and 18F. The published report

includes the detailed process by which new CPS reference sequences, labelled 33F2, 24F2 and 18F, were made available in the references.fasta file located in PneumoKITy CTVdb folder on GitHub.

Updating the Pneumokity CTVdb to add the new 24F2 and 33F2 sequences implies that these serotypes can be interpreted correctly based on the database. Pneumokity thus, has advantages over other WGS pipelines that may have not yet incorporated these refinements.

- Let's examine the PneumoKITy output of S674 as an example:

{674 PneumoKITy serotyping result report

SEROTYPING RESULTS

Stage 1 screen results: 24F/24B

Stage 1 category: type

Stage 1 top hits: {'24F2': 99.6, '24B': 88.4, '24F': 86.9, '24A': 70.9, '07B': 56.7}

Stage 1 max kmer percentage: 99.6

Stage 1 median multiplicity for top hit % (fastq only): 1

Stage 1 Estimated abundance of mix (%) (if mixed and fastq only): None

Analysed in PneumoKITy Stage 1 only

Predicted serotype result: 24F/24B

Result RAG status: GREEN (Analysis passed)}

In this isolate, the predicted serotype was 24F/24B. The percentage identity was the highest for serotype 24F2. The updated PneumoKITy database and the very high max k-mer percentage assigned to serotype 24F2 increase confidence in the serotype assignment by the tool.

Modification to the manuscript:

Table 1 presents PneumoKITy predicted serotypes based on the maximum K-mer percentage obtained, so no modifications will be included in the table. However, the result was reformulated in the text as follows:

Page 12, Lines 273-275: {In all isolates that were assigned to serogroup 24 by Pathogenwatch (n=3), PneumoKITy consistently identified serotype 24F2 as the one with the highest percentage identity.}

29. Table 1. Regarding S650 conventional result 8F. It is a typo?

Response: Thank you for pointing this out. The multiplex PCR assay includes primers for the identification of serogroup 8 and can't distinguish between the serotypes within serogroup 8. This mistake was corrected in Table 1.

30. Table 1. Regarding 19A-II in S685 is a subtype from serotype 19A, since the manuscript just refers to serotypes, just 19A serotype should be indicated.

Response: Serotype 19A-II was replaced by 19A in Table 1.

31. Table 2. For a better understanding and rapid comparison of resistance profiles, please highlight in some way the resistances observed, more than bold.

Response: Discrepancies in results were color-coded as follows:

- Green fill: All WGS results were discordant with AST results.
- Pink fill: One WGS technique was discordant with other WGS techniques or with AST.

However, we are not sure if the colors in Table 2 can be published this way (this depends on the journal's specific formatting guidelines for tables in the final publication)

A positive aspect is that the readers can refer to text ("Antimicrobial Resistance" subsection) to identify the discordant samples/antibiotics already mentioned there, to locate them in Table 2.

The following information was added to the legend of Table 2 as follows:

{The cells denoted by the "--" sign indicate that the isolate was not processed by ONT_V14. Cells colored in green highlight discordance in results between AST and all WGS techniques. Cells colored in pink highlight discordance in results among WGS techniques.}

32. Table 2. Please, add COT abbreviation.

Response: We removed COT from Table 2 and replaced it with Trimethoprim/ Sulfamethoxazole.

33. Table 2. Treatment of pneumococcal infections usually include beta-lactam antibiotics and fluoroquinolones. ¿Why to include such antibiotics in table 2 and not beta-lactam or fluoroquinolones?

Response: The results included in Table 2 are obtained from pathogenwatch which utilizes the PAARSNP software to infer the presence or absence of resistance genetic markers. On the other hand, beta-lactam resistance is predicted by pathogenwatch based on sequence signatures in the transpeptidase domains (TPDs) of three penicillin-binding proteins: pbp1a, pbp2b, and pbp2x. Hence, we opted to include beta lactam resistance results in a separate table (Table S5).

Based on your recommendation we have included fluoroquinolones resistance results in Table 2.

Moreover, the results of fluoroquinolones resistance were included in the manuscript as follows:

Page 13, Lines 292-297: {The concordance in fluoroquinolones resistance between AST and Illumina or ONT_V14 was 93% (25/27). S678 and S686 were predicted sensitive to fluoroquinolones by all WGS techniques, but intermediate or resistant by AST. Interestingly, ONT_V10 predicted the presence of resistance markers against fluoroquinolones in S645 and S647, which were determined susceptible by AST. ONT_V14 revealed the lack of fluoroquinolones resistance markers in both isolates.}

34. Table 2. Why to include Trimethoprim and Sulfamethoxazole genetic resistance per separate? The manuscript should benefit from including both as a combination, since they are only administered to patients in a combined way.

Response: Trimethoprim and Sulfamethoxazole genetic resistance is now included as a combination in Table 2.

Reviewer #5 (Comments for the Author):

In this paper authors compare the Illumina and ONT systems for S. pneumoniae sequencing (27 isolates) and evaluate their performance for serotyping, MLST and antimicrobial resistance predictions.

I find this manuscript interesting to read and the results are certainly valuable to the research community. But many improvements are needed.

Response: Thank you for your positive feedback on the manuscript. We appreciate your recognition of the results' value and are grateful for your insightful comments, which helped in enhancing the quality of the paper.

1. Serotyping failed for 6 out of 27 isolates by Multiplex PCR (labeled as NT). For these NT isolates, authors should use the gold standard Quellung reaction for typing or repeated PCR to resolve serotypes for those. Otherwise, you can't assume WGS typing results for those "NT" isolates are correct (lack of reference results).

The calculation in this sentence seems wrong, "The serogroups of 24/27 (89 %) isolates predicted by WGS were concordant with multiplex PCR (Table 1)."; I believe only 21 out of 27 (77.8%) were concordant (6 as NT by PCR). Please either perform confirmatory typing tests for NT isolates or correct the % concordance.

Response: Based on your recommendation, typing by multiplex PCR was repeated for the six mentioned isolates. Below are the results of the experiment:

Isolate	Serotype (PCR)
645	23A
647	18
664	NT
674	24A/24B/24F
676	24A/24B/24F
685	19A

S664 had an inconclusive result, showing only a very faint band at the serogroup 24 position on gel. Thus, we kept it assigned as non-typeable.

Table 1 was updated with the above results. Moreover, the serotyping results in the manuscript were updated as follows:

Page 12, Lines 270-272: {Sample S664 (1/27) was NT by PCR, and therefore, not included in the concordance analysis between PCR and WGS. The serogroups of 23/26 (88 %) isolates predicted by WGS were concordant with multiplex PCR (**Table 1**).}

2. How many samples were barcoded and multiplexed, loaded on a single flow cell? Is it one sample per flow cell? Averaging how many reads for each sample? You should specify this in the Methods (or include a supp table).

Response: We already included our approach to determining the number of isolates per ONT flow cell in the “Materials and Methods” section as follows:

Page 8, Lines 189-190: {The total combined genome size of the bacterial samples loaded into each R9.4.1 flow cell (FLO-MIN106) was limited to 35 million base pairs.}

Explanation: The total combined genome size of the bacterial samples loaded into an R9.4.1 flow cell should not exceed 35 million base pairs (bp). This limit was established through previous in-house experiments, which concluded that exceeding this combined genome size can negatively impact the total reads output.

Unlike Illumina, we could not find a calculator that accounts for genome size, expected coverage and flow cell data output (Gb) to determine the number of isolates/run.

Here is an example of the bacterial isolates included in one of our ONT sequencing runs:

Sample	Organism	Genome size (bp)
S655	Streptococcus pneumoniae	2,200,000
S660	Streptococcus pneumoniae	2,200,000
S670	Streptococcus pneumoniae	2,200,000
S666	Streptococcus pneumoniae	2,200,000
S667	Streptococcus pneumoniae	2,200,000
S669	Streptococcus pneumoniae	2,200,000
S678	Streptococcus pneumoniae	2,200,000
S682	Streptococcus pneumoniae	2,200,000
S683	Streptococcus pneumoniae	2,200,000
S686	Streptococcus pneumoniae	2,200,000
S688	Streptococcus pneumoniae	2,200,000
PSA	Pseudomonas aeruginosa	7,500,000
Total combined genome size		31,700,000

In the above case, the total combined genome size (31,700,000 bp) was lower than the chosen cutoff (35,000,000 bp).

Of note, S655 was identified in downstream analysis as *Streptococcus oralis*, with a genome size of 2,322,699 bp. S667 was identified as *Brevibacillus invocatus* with a genome size of 6,502,171 bp.

3. Roughly how much cost per sample (including reagent and flow cell)? How much is the cost per sample for Illumina? Authors claim ONT has much lower cost, but data support is necessary (please include a comparison table).

Response: Considering only reagents and flow cells, the cost per sample at our lab is 103.4\$ for Illumina sequencing and 54.06\$ for ONT. Please check the table included in the response to comment 6 by reviewer 4 for the detailed calculation of these costs.

The following reference to a study by Boostrom et al. was added to the introduction to support our claim that ONT has much lower cost, especially when considering the prices of sequencing devices:

Page 5, Lines 126-130: {However, Illumina entails high costs of sample processing, generates fragmented genomes and lacks the ability to resolve highly repetitive genomic regions (12). Boostrom et al. estimated the start-up costs for ONT and Illumina equipment and consumables capable of generating 48 isolates (13). Based on UK prices, the cost was estimated to be £ 11,369 for ONT/Mk1c and £ 107,117 for Illumina.}

4. It looks like ONT is worse for MLST (SNP typing) compared to Illumina due to higher error rates. I believe ONT has strength in resolving recombination events such as AMR gene cassettes, and resistant plasmid transfers. Have authors found evidence that ONT is better at this (any data)?

Response: We did not conduct this kind of in-depth AMR analysis. Although plasmids can play an important role in acquiring drug resistance, they are not the primary mechanism for AMR in *S. pneumoniae*.

5. I think the main advantages of nanopore WGS versus Illumina in clinical settings is its rapid turn-around time due to its real-time sequencing and data analysis features that enable results (typing, AMR, outbreak tracking) within hours, instead of weeks for traditional methods. Do authors plan to streamline the pipeline for real-time prediction? Authors should include this in the discussion.

We did not develop the pipeline. It was provided to us by a developer who made it public on GitHub (<https://github.com/avantonder/assembleBAC-ONT>). Hence, we can't modify it to include real-time prediction.

Library preparation by ONT typically takes between 1.5 and 2.5 hours, depending on the experience of the laboratory technician. Thus, sequencing can be performed on the same day as receiving the sample. With the real-time signaling feature of the MinKnow software, data is generated simultaneously during sequencing. Hence, when enough reads are obtained to produce a sufficient sequencing depth (the cutoff is 30x and above at our lab), AMR and serotyping

results can be generated confidently. This is demonstrated in this work, in which isolates at a sequencing depth of 37X, 39.9X, 41X ext. had high concordance with Illumina and AST data. The V14 chemistry from ONT, which replaced the V10 chemistry, further improves sequencing accuracy and decreases the time needed to generate raw reads. Usually, 24 hours of sequencing on the Mk1c or Mk1b devices are enough to produce sufficient reads for analysis. After this period, the lab personnel can retrieve the data from the device and start the analysis using the pipeline. So the time from sample to results is still much lower compared to classic testing.

Minor comments:

6. Line 130-32, what does "The lack of need for sample batching" mean? Does it imply that authors suggest running 1 sample per flow cell?

Response: Thank you for this comment. The sentence indeed implied that one sample is loaded to an ONT flow cell, which is not the case (this is not cost-effective). It was modified to better reflect that a lower number of samples can be included in a flow cell compared to Illumina:

Page, Line: {The reduced need for large-scale sample batching, compared to Illumina, contributes to the promising potential of ONT as a diagnostic technique that can be adopted in the clinical setting.}

References:

- [1] I. Boostrom, E. A. R. Portal, O. B. Spiller, T. R. Walsh, and K. Sands, "Comparing Long-Read Assemblers to Explore the Potential of a Sustainable Low-Cost, Low-Infrastructure Approach to Sequence Antimicrobial Resistant Bacteria With Oxford Nanopore Sequencing," *Front. Microbiol.*, vol. 13, 2022, doi: 10.3389/fmicb.2022.796465.
- [2] R. R. Wick and K. E. Holt, "Benchmarking of long-read assemblers for prokaryote whole genome sequencing.," *F1000Research*, vol. 8, p. 2138, 2019, doi: 10.12688/f1000research.21782.4.
- [3] N. Lermينياux, K. Fakharuddin, M. R. Mulvey, and L. Mataseje, "Do we still need Illumina sequencing data? Evaluating Oxford Nanopore Technologies R10.4.1 flow cells and the Rapid v14 library prep kit for Gram negative bacteria whole genome assemblies.," *Can. J. Microbiol.*, Feb. 2024, doi: 10.1139/cjm-2023-0175.
- [4] J. Linde *et al.*, "Comparison of Illumina and Oxford Nanopore Technology for genome analysis of *Francisella tularensis*, *Bacillus anthracis*, and *Brucella suis*.,," *BMC Genomics*, vol. 24, no. 1, p. 258, May 2023, doi: 10.1186/s12864-023-09343-z.
- [5] G. Kapatai *et al.*, "Whole genome sequencing of *Streptococcus pneumoniae*: development, evaluation and verification of targets for serogroup and serotype prediction using an automated pipeline.," *PeerJ*, vol. 4, p. e2477, 2016, doi: 10.7717/peerj.2477.
- [6] C. L. Sheppard *et al.*, "PneumoKITy: A fast, flexible, specific, and sensitive tool for *Streptococcus pneumoniae* serotype screening and mixed serotype detection from genome sequence data.," *Microb. Genomics*, vol. 8, no. 12, Dec. 2022, doi: 10.1099/mgen.0.000904.

Re: Spectrum01294-24R2 (Comparison of Illumina and Oxford Nanopore Technology systems for the genomic characterization of *Streptococcus pneumoniae*)

Dear Prof. Ghassan Dbaibo:

Your manuscript has been accepted, and I am forwarding it to the ASM production staff for publication. Your paper will first be checked to make sure all elements meet the technical requirements. ASM staff will contact you if anything needs to be revised before copyediting and production can begin. Otherwise, you will be notified when your proofs are ready to be viewed.

Sincerely,
Sophia Georghiou
Editor
Microbiology Spectrum

Reviewer #4 (Public repository details (Required)):

Raw fastq files should be deposited in a public repository like ENA or SRA.

Reviewer #5 (Comments for the Author):

The authors have done a great job addressing my questions. I have no further comments.